# Polar discontinuities, emergent conductivity, and critical twist-angle-dependent behaviour at wafer-bonded ferroelectric interfaces

Andrew Rogers [1,12], Kristina Holsgrove[1,12] ✉, Nils A. Schäfer[2], Boris Koppitz [3], Conor J. McCluskey[1], Shivani Yedama[3], Ronan Lynch[1], Keelan Sloan [1], Barry Porter[1], Adam Sykes[1], Alex Catalan Daniels[1], Romualdo S. Silva Jr. [4], Flavio Y. Bruno [4], Sam D. Seddon [3], Haidong Lu [5], Michael Ruesing [6], Christa Fink [2], Philipp Fahler-Muenzer [7], Sarah Fearn [8], Sandrine E. M. Heutz [8], Marios Hadjimichael[7], Quentin M. Ramasse [9,10], Marin Alexe [7], Amit Kumar [1], Raymond G. P. McQuaid [1], Alexei Gruverman[5], Simone Sanna [2], Lukas M. Eng [3,11] & J. Marty Gregg [1] ✉

Probing novel properties, arising from twisted interfaces, has traditionally relied on the stacking of exfoliated two-dimensional materials and the spontaneous formation of van der Waals bonds. So far, investigations involving intimate covalent or ionic bonds have not been a focus. Yet, we show here that an established technique, involving thermocompressional wafer bonding, works well for creating twisted non-van der Waals interfaces. We have successfully bonded z-cut lithium niobate single crystals to create ferroelectric oxide interfaces with strong polar discontinuities and have mapped the associated emergent interfacial conductivity. In some instances, a dramatic change in microstructure occurs, involving local dipolar switching. A twist-induced collapse in the capability of the system to effec8tively screen interfacial bound charge is implied. Importantly, this only occurs around specific moiré twist angles with sparse coincident lattices and associated short-range aperiodicity. In quasicrystals, aperiodicity is known to induce pseudo-bandgaps and we suspect a similar phenomenon here.

When two materials' surfaces are brought together, new properties, beyond those of either one in isolation, can spontaneously develop. Sometimes this is obvious; for example, if one material is magnetoelastic, while another is piezoelectric, then intimate contact facilitates strain transfer and a resulting overall magnetoelectric response, which would not otherwise be present[1,2]. Less obvious phenomena occur when the interface itself develops its own unique properties. For example, two insulators bonded together may induce strong two-dimensional (2D) or pseudo 2D conductivity and even superconductivity where they abut[3,4]. Equally, interfaces in non-magnetic systems can themselves be magnetic[5,6] and, in non-polar systems, ferrielectric (in $CaTiO_3$)[7], or even flexoferroelectric (in $SrTiO_3$)[8].

If conjoined, with a relative twist in lattice orientation about an axis perpendicular to the bonded interface, further intriguing behaviour may emerge, such that interfacial properties can be tuned solely through twist angle variation[9]. In graphene, for example, two layers

stacked together with a relative twist of ~1.1° will induce superconductivity[10], but further minor twist adjustments, even of the order of 0.1°, collapse the superconducting state. Twist-induced correlated insulating behaviour can equally be made-to-order[11]. Furthermore, magic-angle twisting of two van der Waals (vdW) oxide layers may lead to the propagation of canalised topological low-loss polaritons along that interface[12].

To date, this kind of twistronics has almost universally been the preserve of layered vdW materials, where twisted stacks, supporting relatively weak bonds, can effectively be created at room temperature. Twistronic investigations in materials, in which more intimate covalent or ionic bonds are needed, have lagged behind. Nevertheless, recent progress has been made, capitalising on the inclusion of sacrificial water-soluble buffer layers in epitaxial oxide heterostructures, to allow free-standing thin films to be detached, lifted, and then manually placed on top of each other[13–15]. Results have been exciting. Noteworthy recent studies[16,17] have revealed new flexoelectrically-driven dipolar textures, associated with modulations in strain that arise from periodic variations in lattice mismatch across the twisted interface (moiré patterns).

We herein report that straightforward, relatively high temperature thermocompression processing of bulk single-crystal oxide wafers can also be extremely effective in producing extended, clean and well-bonded twisted interfaces that are clearly not reliant on any vdW interactions. Moreover, we show that such a bonding methodology gives ready access to emergent and twist-dependent interfacial physics. We focus on z-cut ferroelectric lithium niobate (LNO). Although LNO is well-studied and bonding of such crystals has historically been used to form optical waveguides[18,19], no reports have been published on the electrical properties of bonded interfaces, or on any associated influence of relative lattice twist.

We have specifically targeted the formation of interfacial polarisation discontinuities, as it is now well established that they frequently induce emergent conductivity in both epitaxial heterostructure interfaces[3] and in charged ferroelectric domain walls[20], as a result of strong potential variation and highly localised band-bending[21]. A key motivation in our research was to map and understand the effects that twists might have on such interfacial conductivity. In this context, LNO was chosen for several reasons: firstly, it is a uniaxial ferroelectric and hence head-to-head (H2H) or tail-to-tail (T2T) polarisation discontinuities cannot be readily obviated through the formation of 90° closure domains, or through local dipolar rotation; secondly, LNO has a Curie Temperature (~1140 °C) close to its melting temperature (~1240 °C)[22], and hence robust thermal bonding at or above typical sintering temperatures can be achieved entirely within the ferroelectric state. Uncontrolled reconstruction of the polar domains, that would inevitably occur on thermal cycling through the ferroelectric-paraelectric phase transition, is therefore avoided. We emphasise that these features of LNO are rather special. In common perovskite ferroelectrics, such as BaTiO$_3$ or Pb(Zr,Ti)O$_3$, multiple equivalent domain orientations could readily allow polarisation patterns which would avoid the formation of an interfacial polar discontinuity. Equally, for such systems, intimate thermocompressive bonding within the ferroelectric state would be unlikely, as Curie Temperatures are dramatically lower than their melting temperatures.

Our results show the emergence of two distinct behaviours at the bonded interfaces. In most cases, preset H2H polar configurations survive the bonding process, polar discontinuities are maintained, and pseudo 2D interfacial conductivity arises. In a few cases, which manifest only around distinct twist angles, the polarisation close to the interface reverses its orientation, becoming T2T. Two untwisted conventional H2H domain walls then form nearby. They are electrically conducting and are therefore expected to facilitate effective screening in the system. We suggest that this radical microstructural change results from twist-induced short-range aperiodicity related to a sparse interfacial coincident lattice, an induced pseudogap formation in the local electronic band structure (as seen in quasicrystals) and the resulting electric fields from positive bound charge that cannot then be effectively screened. We note that the link between dipolar inversion layers and unscreened electric fields has previously been made for exposed (0001) surfaces in furnace-treated LNO bulk crystals[23–28]. However, in these prior studies, completely different origins of unscreened charge were at play, involving extensive ionic diffusion in an environment with strong chemical concentration gradients.

## Results

### Fabrication and structural characterization of bonded LNO interfaces

We initially bonded wafers with (0001) surfaces (z-cut), arranged in a H2H configuration, aligned such that there were no deliberate relative twists in the lattices across the interface. Bonding success was achieved by heating in air, in a furnace, at 1100 °C (for 20 h), with ~20 kPa of pressure applied perpendicular to the wafer (0001) surfaces (details in Fig. S1). After cooling, cross sections of the interface were cut and examined using (scanning) transmission electron microscopy ((S)TEM). As can be seen in Fig. 1, the resulting interfaces were clean and atomically sharp, as evidenced by strong continuity in the Nb-atomic columns through the interface (Fig. 1C, D). The H2H polar discontinuity, as configured in the stacked unbonded crystals at room temperature, was also maintained after the bonding process (evidence from differential phase contrast [DPC], where the electron beam deflection registered on segmented annular detectors directly reflects the local polarisation directions, Fig. 1B). Digital dark-field imaging (Fig. 1E–G), using fine apertures to select specific periodicities in the Fast-Fourier Transforms of the high-angular dark field scanning transmission electron microscopy (STEM) images (Fig. 1E), showed that the LNO unit cell is progressively compressed along the [0001] direction, on approaching the interface from the bulk crystal on either side. At its maximum, at the interface, the compressive strain of the unit cells along [0001] reaches ~5% (Fig. 1H); data used in Fig. 1H was obtained by averaging intercolumnar separations between Nb columns, as a function of distance from the bonded interface. Such compression has additionally been confirmed using both Geometric Phase Analysis and Atomap software (supplementary information, Fig. S2). We note parenthetically that some of our most recent data suggest this local strain state changes its sense (from compression to extension) when untwisted T2T, as opposed to H2H, polar discontinuities are considered.

Examination of polished cross-sections, using piezoresponse force microscopy (PFM), confirmed the H2H polarisation discontinuity at the bonded interface (Fig. 2A–C), consolidating inferences from the DPC contrast, discussed above (Fig. 1B).

To investigate the 3D morphology of the interface, we performed Cherenkov second harmonic generation (CHSG) imaging, used successfully in previous work to map domain wall loci over macroscopic length scales[29–35]. In this technique, only boundaries generate second harmonic contrast. Fig. 2F shows a CHSG-generated 3D reconstruction of a sample with a H2H bonded interface, which appears to have a continuous planar structure throughout the entirety of the imaged region.

### Electrical conduction along the bonded interfaces

Bulk LNO is a strongly insulating material with a ~4 eV optical bandgap[36]. However, it is well established that domain walls in LNO, separating regions of uniform polarisation, can be strongly conducting[20,37–39], provided they support polarisation discontinuities. We therefore expected that the thermocompressively bonded H2H interface might also be electrically conducting. As can be seen in Fig. 2D, conducting atomic force microscopy (cAFM) confirms it, with clear current signals arising only at the interface itself. cAFM contrast

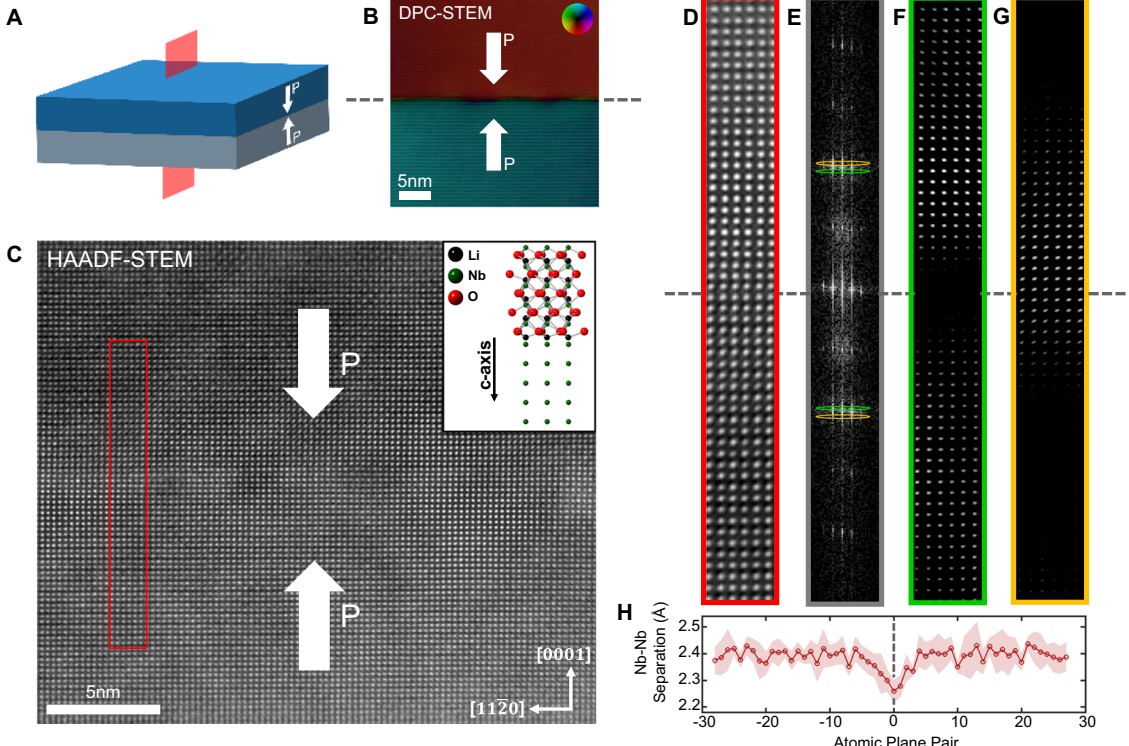

**Fig. 1 | Scanning transmission electron microscopy (STEM) analysis of bonded LNO interface. A** Schematic of bonded H2H sample. **B** DPC-STEM on bonded interface region; the centre of the interface is marked with grey dashed lines. **C** HAADF-STEM of the bonded interface displaying niobium atomic columns viewed along [1̄100]; a schematic of the relative positions of the different chemical species in the LNO structure, as viewed along [1̄100], is shown by the "ball and stick" representation in the upper part of the inset. The lower part of the inset is the same projection in which only the Nb atoms are rendered. This can be directly compared with the HAADF image. **D** Higher magnification HAADF-STEM image from the red boxed region in (**C**). **E** Fast Fourier Transform (FFT) of (**D**) showing apertures moved through the 0006 Bragg peaks. **F, G** Inverse FFTs performed for the green, and orange apertures in (**E**), showing the difference in Nb-Nb periodicity adjacent to and away from the bonded interface. **H** Nb-Nb spacing measured along the c-axis within (**D**); positive and negative directions refer to Nb-Nb pairs in the top crystal and bottom crystal, respectively, moving away from the interface. Shading indicates calculated standard deviation obtained by averaging across multiple atomic rows.

suggests that conductivity is not entirely uniform. However, we suspect this is due to imperfect polishing and spatially intermittent electrical contact between tip and sample during scanning. Certainly, better polished twisted samples (discussed later) showed more continuous interfacial cAFM contrast. The integrated current, across all pixels equidistant from the bonded interface in the cAFM scan region illustrated, shows the strong spatial confinement of conduction (Fig. 2G), reminiscent of prior measurements done on 2D electron gases (2DEGs)[40].

To understand more about the origin of this conduction, we probed for evidence of stoichiometric variation, and the possibility of associated electronic defect states close to the interface. Information from Electron Energy Loss Spectroscopy (EELS) suggested minimal interfacial oxygen loss (illustrated in Fig. 3A–C and Figs. S3 and S4). However, in many systems, even modest deoxygenation can cause conductivity to arise. We therefore performed an additional experiment in which a thin aluminium film was deposited onto the surface of an unbonded, isolated z-cut LNO single-crystal under an argon atmosphere at 250 °C, following the procedure described in Ref. 41. Typically, the oxidation of Al into $AlO_x$, via a redox reaction, depletes oxygen from the underlying crystal, forming an oxygen-deficient surface layer. In materials such as strontium titanate ($SrTiO_3$) and potassium tantalate ($KTaO_3$), the process induces the formation of metallic interfacial 2DEGs[42–47]. However, in the case of LNO, the resistance remained high, despite the presumed local removal of oxygen: it remained at least four orders of magnitude greater than that of a KTO sample processed under identical conditions (Fig. S5) and was

comparable to that of a control sample grown on an insulating $LaAlO_3$ substrate. This suggests that oxygen vacancies, even if present, are not responsible for the observed emergent conductivity at the bonded interfaces.

To gain further insight, Density Functional Theory (DFT) was used to understand the electronic structure changes when two z-cut LNO crystals, initially apart, are subsequently intimately bonded. Figure 3D, E shows the calculated band structures from a DFT supercell containing both a H2H and T2T interface. The supercell models a bonded structure consisting of two z-cut LNO layers terminated by the thermodynamically stable c+ surface, according to Ref. 48. The atomic structure, as well as the computational details, are presented in the Fig. S6. The contributions of all atoms within a span of 10 Å, centred at the H2H interface, are color-coded. The oxygen *p* orbitals (Fig. 3D) dominate the band structure (in particular, the valence states), while contributions from the niobium *d* bands (Fig. 3E) are comparatively minor (and localized at the conduction states). Contributions from lithium *s* states and other niobium and oxygen orbitals are not shown, as their share is negligible. We furthermore observe that the polarisation discontinuity-induced shift of the electronic bands raises the valence states at the H2H interface slightly above the Fermi level, rendering the system (semi) metallic. This effect is clearly visible in the real space representation of the density of states (DOS) along the c-direction, shown in Fig. 3F. The polarisation discontinuity causes an upward shift of the electronic states at the H2H interface, and concurrently, a downward shift at the T2T interface.

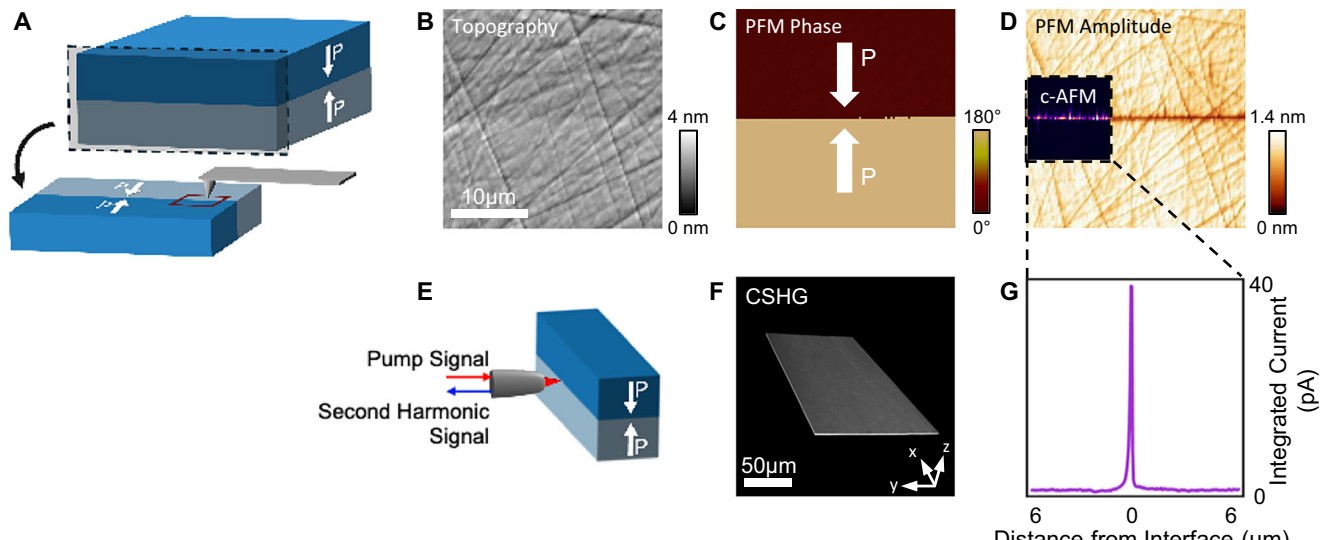

**Fig. 2 | Abutting head-to-head (H2H) domains at the interface of two thermo-compressively bonded LNO crystals. A** Schematic of H2H sample processing for scanning probe microscopy imaging. Sample is cut and polished to allow for AFM analysis of the interface. **B–D** AFM topography, PFM phase, PFM amplitude and c-AFM of bonded region (inset). c-AFM (inset in **D**) was obtained with bias of −9.5 V applied to the base of the sample. **E** Schematic of second-harmonic microscopy for imaging the interface. **F** 3D Cherenkov second-harmonic generation (CSHG) image of a region of the H2H bonded interface. **G** Plot of integrated current against distance from the interface taken from the c-AFM data (inset in **D**).

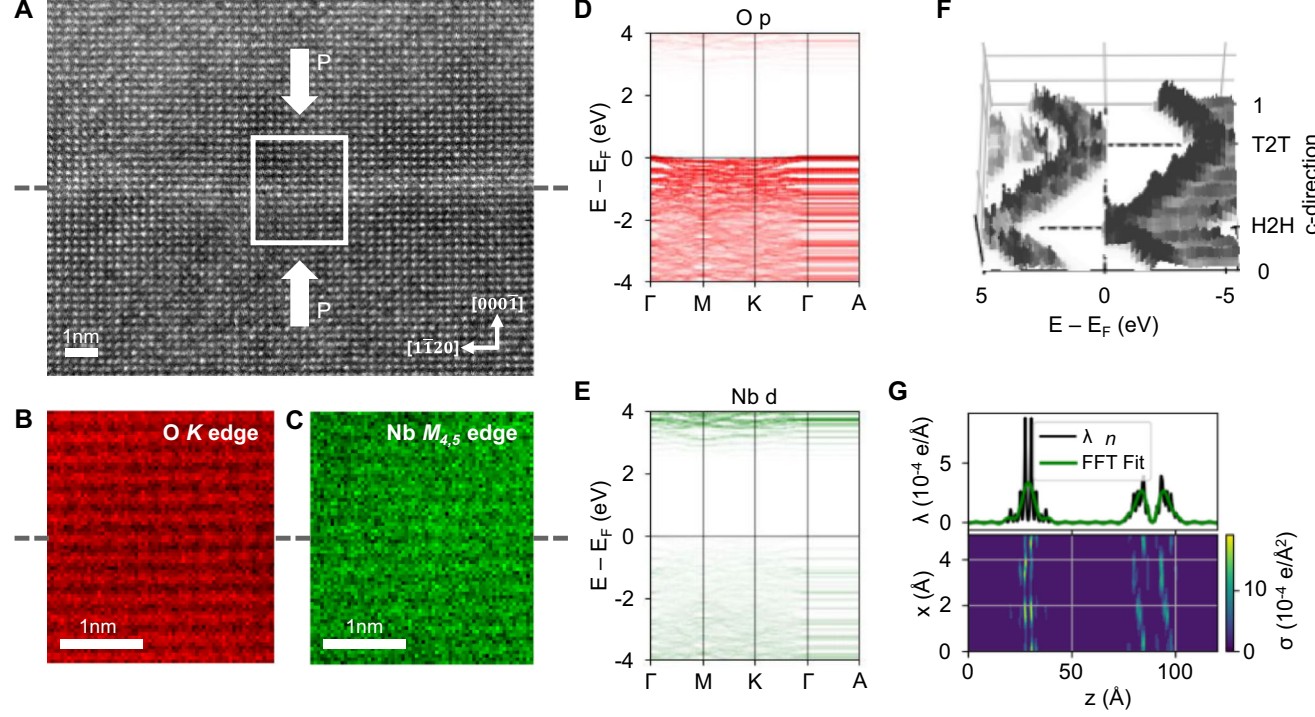

**Fig. 3 | Electronic structure of LNO bonded interface. A** MAADF-STEM image of a H2H interface. **B** EELS signal map of O K edge and **C** Nb $M_{4,5}$ edge from the white boxed region in (**A**). Modelled electronic band structures displaying the **D** O p orbitals and **E** Nb d orbitals. **F** Real space representation of the calculated DOS along the c-axis. **G** The charge density averaged along the c-axis.

Correspondingly, the valence states at the H2H and the conduction states at the T2T are shifted to the Fermi level (eventually crossing it in the case of H2H), creating a charge carrier gas. Figure 3G shows the charge density, averaged along the c-axis (solid black line) as well as its FFT fit, ruling out the rapid charge oscillations (solid green line). The FFT fit is, to a very good approximation, the Jacobi elliptical function, as derived within Ginzburg-Landau-Devonshire theory to describe the charge profile[49]. As shown in Fig. 3G, the free

charge carriers are predicted to be more localized at the H2H, than at the T2T interface (reflecting experimental observations, insofar as a distinct pseudo-2D conducting sheet is seen using cAFM for H2H bonded interfaces only; T2T interfaces have no contrast in cAFM, but do show macroscopic conduction greater than the bulk crystal, illustrated in Figs. S7 and S8). This is due to the fact that the H2H and T2T regions are structurally (and chemically) different. The presence of strongly localized free charge carriers at the Fermi energy (see lower

part of Fig. 3G) further suggests the formation of a 2D conductive region.

We note that the predicted behaviour at a bonded H2H or T2T interface differs from plain charged domain walls (CDW) in bulk LNO. At a H2H CDW, the conduction bands cross the Fermi energy, and at a T2T CDW, the valence bands cross[50]. The difference between a bonded H2H or T2T interface, and a plain H2H or T2T CDW, is the presence of a surface relaxation at the bonded structure, which we model as the interface between two thermodynamically stable surfaces. The surface relaxation induces a strong local dipole moment opposite to the spontaneous polarisation, in agreement with the predicted shift of the electronic states at the interface. The present calculations show that the atomistic modelling of the bonded structure is indeed possible and, furthermore, that the polarisation-induced band bending at the bonded interface may locally create a charge carrier gas, as in bulk domain walls. However, the experimental knowledge of the atomic structure of the bonded interface, including lithium atom positions, is crucial for a fully quantitative description of the electronic structure and this experimental information could not be reliably obtained (see Fig. S9).

## Deliberate introduction of a twist angle

We next discuss the influence of introducing a relative twist between the two single-crystal LNO wafers, prior to bonding (illustrated in Fig. 4A). LNO belongs to the trigonal crystal system and so has a three-fold rotational symmetry about its c-axis. Nevertheless, the lattice point distribution for hexagonal and trigonal unit cells is the same. Somewhat counterintuitively, therefore, the trigonal unit cell enables perfect coincidence of interfacial lattice points, for crystals abutting on (0001) surfaces, at twist angles of multiples of 60° (not 120°). HAADF-STEM characterization and electron diffraction, of the interface region in a ~60° twisted H2H bonded system (Fig. 4B and S10), confirms this, insofar as the niobium columns in each of the wafers meet coherently. The twist is manifest in the different relative orientations of the niobium columns, on moving away from this largely coherent interface. PFM phase information (Fig. 4C) confirms that the H2H polar discontinuity is maintained, and cAFM mapping (Fig. 4D) shows similarly enhanced conductivity to that observed in untwisted bonded wafers. We place no significance on the difference in the magnitude of measured currents in Figs. 4D and 2G, as this varies strongly with tip quality and surface polish conditions.

While the ~60° twist clearly represents a special case, other twist angles also generate clean bonds and generally maintain H2H polar discontinuities; the ~3° and ~70° cases shown in the supplementary information (Fig. S11) are good examples. However, at several specific angles, distinctly different behaviour emerges: in our study, twist angles of ~14°, ~21°, and their moiré equivalents (~74° for example).

To illustrate, let us consider observations made for bonded LNO with a twist angle of ~21°, for which the experimental moiré electron diffraction pattern acquired along the c-axis is shown in Fig. S12C; it can be compared to the simulated electron diffraction patterns of two LNO crystals twisted by 21° and superposed (in Fig. S12F). Figure 5 shows that dramatic changes have clearly developed during thermo-compression processing, resulting in a microstructure unseen in any of the untwisted samples we have made to date (and unseen in most of the twisted samples too). The polarisation in the region adjacent to the bonded interface has reversed its orientation, such that the polar discontinuity has become T2T, rather than the initial H2H, as set at room temperature prior to bonding (Fig. 5C). Polar inversion has not progressed beyond some ~15 μm from the interface and so, inevitably, charged domain walls (H2H) have also formed within each of the single crystals on either side of the bond. Importantly, these H2H conventional domain walls are twist-free and are electrically conducting (Fig. 5D); small angles in wall kinks are commensurate with such conduction, as they indicate reasonably effective screening[51].

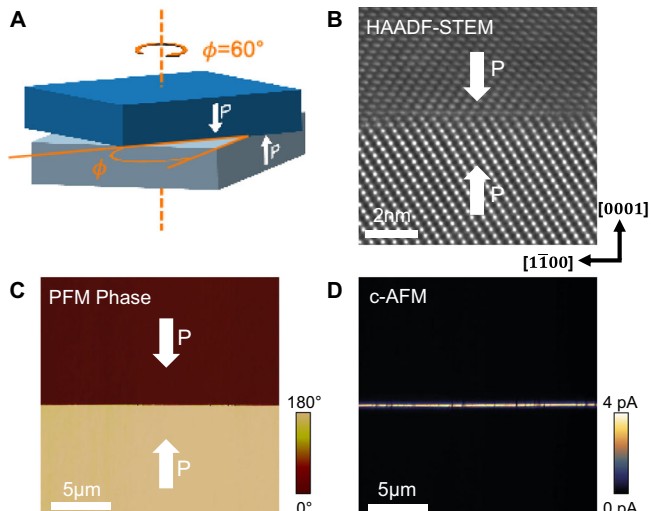

**Fig. 4 | H2H bonded sample with ~60° interfacial lattice twist. A** Schematic of H2H bonded sample with a ~60° twist. **B** Atomic resolution HAADF-STEM showing the niobium columns in the vicinity of the ~60° twisted H2H interface. **C** Lateral PFM phase map of bonded interface. **D** Conductive-AFM map of the bonded interface, with bias of −9.5 V applied to the sample base.

The manifestation of the spontaneous strain for "up" and "down" polarisation orientations in LNO is the same and so the local reversal of polarisation described above could not have been induced by uniform stress. In principle, a flexoelectric field (resulting from a stress gradient) could have been responsible for the local switching. However, we see little evidence that strain gradients perpendicular to the bonded interfaces vary with twist angle. The most likely reason for the local polar switching observed for the ~21° twisted case (and for the ~14° and ~74° cases) is therefore the presence of a conventional electric field, greater than the coercive field for LNO, with flux emanating from the interface itself. Importantly, the sense of polar inversion (H2H changing to T2T) is consistent with fields produced by poorly screened positive interfacial bound charges. In the majority of the H2H bonded samples, where no inversion has been seen, bound charge screening is enabled by the emergent interfacial conductivity. We therefore hypothesize that at specific angles of twist (~14°, ~21°, ~74°), the electronic band structure at the interface changes to shut down conduction and prevent the system from effectively screening the positive bound charge. This creates the switching field, which is then destroyed by the domain inversion process and the formation of two H2H conducting domain walls.

The specific angles, at which polar inversion is seen in our work, correlate well with unusual ranges in twist angle for hexagonal lattice meshes[52] at which coincident lattice spacings dramatically increase (Fig. S13) and generate coincident site lattice band gaps (Fig. S14), as well as short-range aperiodicity and disorder. In quasicrystals, aperiodicity is known to be accompanied by the formation of electronic pseudo band-gaps[53,54] and a dramatic increase in resistance[55–57]. We strongly suspect similar aperiodicity-related physics for twisted LNO interfaces at the specific twist angles where dipolar inversion is seen. We note that the specific angular ranges at which aperiodicity/disorder emerges depends on the symmetry of the interfacial lattice mesh. We consider hexagonal meshes (in Figs S13 and S14)[52], but other studies have considered other mesh combinations[58,59].

The domain inversions seen in Fig. 5 appear similar to those that often develop at free surfaces in heat-treated single crystals of LiNbO$_3$, LiTaO$_3$, and KTiOPO$_4$[23–28]. In the case of heat-treated z-cut LNO specifically, a domain of opposite polarity always seems to nucleate at the free c+ face. This effect is typically assigned to an out

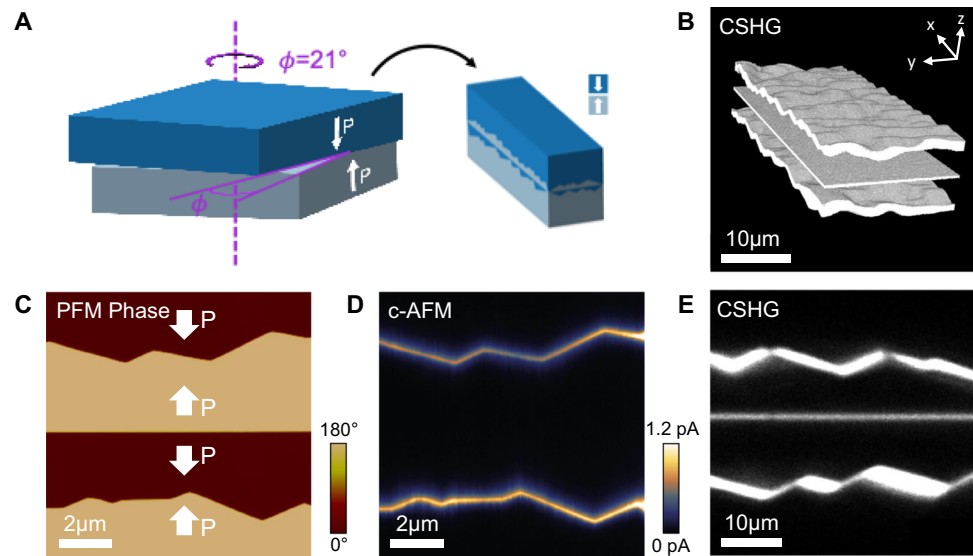

**Fig. 5 | Influence of twist upon interface polarisation state. A** Schematic of the production of a H2H sample with a twist angle ( - 21°). **B** 3D CSHG image showing the presence of three boundaries. **C** Lateral PFM phase of the bonded region showing a microstructural domain inversion layer at interface. **D** c-AFM of bonded region obtained with a bias of −9.5 V applied to the sample base. **E** 2D CSHG image, taken at a different bonded region to (**C** and **D**).

diffusion of $Li_2O$ from the c+ face, resulting in a defect concentration gradient and a built-in unscreened electric field, which can exceed the coercive field of LNO (greatly lowered at high temperature, on the order of $5\,Vcm^{-1}$ at 1100 °C[21]). This mechanism is supported by observations of variations in optical properties when moving away from the c+ face of heat-treated crystals[60,61]. Notably, domain inversions do not occur in regions on the c+ face that are capped with a metallic layer[62,63].

While we are content that such previous studies show that interface-related (free surface-related) polar inversion can occur as a result of unscreened electric fields, we do not think that the inversion in our samples can be attributed to interface-related ionic diffusion. Using Secondary Ion Mass Spectroscopy on a Focused Ion Beam (FIB) microscope, we tracked the normalised $Li^+$ ion concentration signal, as a function of position with respect to the interface, in a case where no polar inversion occurred and in a case where it did occur (Fig. S15). No measurable monotonic spatial variation in $Li^+$ ion concentration was found, as a function of distance away from the interface, in either case. There is therefore nothing to support the notion of any twist-related enhancement of lithium efflux. Rather, it seems most plausible that the electric field responsible for polar inversion arises from unscreened bound charge at the H2H polar discontinuity.

In summary, results from the characterisation of charged bonded LNO wafers suggest that the screening capabilities of interfaces with H2H polar discontinuities are indeed twist-dependent. Effective screening appears to be disrupted by local aperiodicity, which develops at magic angles where the interfacial coincident lattice becomes extremely sparse.

## Methods
### Thermocompressional wafer bonding
The single-crystalline congruent z-cut LNO used in this study was obtained from two suppliers, with no nominal difference in growth or composition (MTI Corporation and Shanghai BonTek Optoelectronic Technology Development Co., Ltd.). Thermocompression bonding was achieved inside a Carbolite RHF1400 furnace. The LNO wafers were cleaned with acetone, isopropanol, and water before bonding. Zirconium oxide crucibles filled with zirconium oxide powder were used to generate ~20 kPa downward pressure to the LNO wafers. The programmed thermal cycle used is given in Fig. S1.

### Scanning probe microscopy
Polarisation imaging was performed using lateral PFM on an Asylum Research MFP-3D Infinity Atomic Force Microscope. For such measurements, a Pt-coated Si tip (Nanosensor PPP-EFM) with a stiffness constant of $2.8\,Nm^{-1}$ and typical resonance frequency of ~70 kHz was used.

Conductive Atomic Force Microscopy (cAFM) was performed on the same Asylum Research MFP-3D, using a boron-doped diamond-coated Si tip (Nanoworld) with a stiffness constant of $80\,Nm^{-1}$ and typical resonance frequency of ~400 kHz. Bonded samples were mounted on a metal disc with conductive silver paste to allow for the application of a bias to the base of the sample.

### Cherenkov second-harmonic generation (CSHG) imaging
CSHG measurements were performed using an inverted Zeiss LSM 980 microscope and a Ti:Sapphire laser (Spectra Physics Insight ×3, 680–1300 nm, 2 W maximum power). All measurements were captured in the back-scattering geometry using a 20× magnification plan-apochromat objective with a 0.8 numerical aperture (NA). The incident laser is linearly polarized and blocked with appropriate colour filters after illumination of the sample. For the measurements in this work, the laser was tuned to a fundamental wavelength of 900 nm. The second harmonic light is detected via a GaAsP photocathode photo multiplier tube. A galvanometer scanner is used to move the laser focus laterally across the sample, while a z-stage allows measurements to be taken at different depths into the sample.

### Focused ion beam processing and electron microscopy imaging
A FIB microscope was used to prepare cross-sectional lamellae for transmission electron microscopy (Tescan-focused ion beam-secondary electron microscope Lyra 3). FIB milling was performed with gallium ions at accelerating voltages of 30 kV (rough milling), 5, 2, and 1 kV (fine polishing). STEM measurements were acquired on a ThermoFisher Scientific Talos F200X instrument operated at a 200 kV accelerating voltage. For DPC imaging, a segmented detector (ThermoFisher) was used to measure deflection of the transmitted electron beam. Atomic-resolution STEM images were acquired on a Nion UltraSTEM 100 operated at 100 kV, with a convergence semiangle of 30 mrad and a probe current of ~30 pA. The collection semiangle range for HAADF-STEM images was 89–195 mrad and MAADF-STEM images

was 47–85 mrad. EELS was acquired on a UHV Enfina EEL spectrometer with a collection semiangle of 22 mrad and energy resolution of 0.4 eV.

## Computational simulations

To explore the bonding characteristics of z-cut $LiNbO_3$, first-principles calculations were carried out using DFT, as implemented in VASP. The projector-augmented wave method, along with the PBEsol functional is applied. The cutoff for the plane wave basis is set at 475 eV. Explicit treatment is given to 1, 13, and 6 valence electrons for Li, Nb, and O, respectively. Structural relaxations are performed using the RMM-DIIS algorithm until the forces acting on the single atoms are smaller than 0.005 eV/Å. The convergence criterion for the self-consistent determination of the electronic charge is reached if total energy changes between consecutive iterations fall below $10^{-6}$ eV. Thereby the energy integration is performed utilising a Γ centred $4 \times 4 \times 1$ k-point mesh.

## Data availability

The data generated in this study have been deposited on the Queen's Research Repository database[64].

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

## Acknowledgements

Mike N. Pionteck at the Justus-Liebig-Universität of Gießen is acknowledged for useful discussion and technical support. Dr Gabriele De Luca and Dr Tchavdar Todorov are acknowledged for helpful discussions. J.M. Gregg acknowledges financial support from the UK Engineering and Physical Sciences Research Council (EP/X027074/; J.M.G.) for the CAMIE Programme Grant. K. Holsgrove acknowledges financial support from SuperSTEM, the UK's National Facility for Aberration Corrected STEM, funded by the Engineering and Physical Sciences Research Council, and the Department of Education and Learning NI through grant USI-205; A.K. and K.H. A. Rogers acknowledges financial support from the UK Department for the Economy (DfE). F.Y. Bruno acknowledges financial support through grant CNS2022-135485 funded by MCIN/AEI/ 10.13039/ 50110001103 and Unión Europea NextGeneration EU/PRTR (FYB). R. McQuaid gratefully acknowledges support by a UKRI Future Leaders Fellowship (Grant Number MR/T043172/1; R.G.P.M.). S.Sanna and L.M. Eng gratefully acknowledge financial support by the Deutsche Forschungsgemeinschaft (DFG) through the DFG research group FOR5044 (grant no. 426703838; https://www.for5044.de; S.S. and L.M.E.). L.M. Eng gratefully acknowledges the financial support by the Deutsche Forschungsgemeinschaft (DFG) through the Würzburg-Dresden Cluster of Excellence on "Complexity and Topology in Quantum Matter"—ct.qmat (EXC 2147; ID 39085490; L.M.E.). A. Gruverman acknowledges support by the UNL Grand Challenges catalyst award "Quantum Approaches Addressing Global Threats". P. Fahler-Muenzer acknowledges funding from the Chancellors Scholarship of the University of Warwick. Calculations for this research were conducted on the Lichtenberg high-performance computer of the TU Darmstadt and at the Höchstleistungrechenzentrum Stuttgart (HLRS). The authors, furthermore, acknowledge the computational resources provided by the HPC Core Facility and the HRZ of the Justus-Liebig-Universität Gießen. We acknowledge the excellent support of the Light Microscopy Facility, a Core Facility of the CMCB Technology Platform at TU Dresden, where the SHG analysis was performed.

## Author contributions

Conceptualization: J.M.G., L.M.E., A.G., Methodology: K.H., C.J.M., Q.M.R., J.M.G., L.M.E., R.G.P.M., A.K., N.A.S., P.F.-M., M.H., M.A., M.R., F.Y.B., S.E.M.H., Investigation: K.H., A.R., C.J.M., N.A.S., C.F., B.K., A.C.D., R.L., B.P., A.S., S.Y., F.Y.B., H.L., Q.M.R., P.F.-M., R.S.S.Jr, K.S., S.F., S.E.M.H., Visualization: K.H., A.R., N.A.S., B.K., Q.M.R., A.C.D., Funding acquisition: J.M.G., L.M.E., K.H., S.S., A.K., S.E.M.H., Project administration: J.M.G., L.M.E., Supervision: J.M.G., L.M.E., S.S., M.H., M.A., F.Y.B., A.G., Writing–original draft: K.H., A.R., J.M.G., N.A.S., C.J.M., S.S., A.G., L.M.E., Writing–review & editing: J.M.G., L.M.E., K.H., A.R., N.A.S., C.J.M., B.K., R.G.P.M., S.D.S., S.S., M.H., M.A., P.F.-M., Q.M.R., A.G., F.Y.B., S.F., S.E.M.H.

## Competing interests

The authors declare no competing interests.

## Additional information

[1]Centre for Quantum Materials and Technologies, School of Mathematics and Physics, Queen's University Belfast, Belfast, UK. [2]Institut für Theoretische Physik and Center for Materials Research (ZfM/LaMa), Justus-Liebig-Universität Gießen, Gießen, Germany. [3]Institute of Applied Physics, TU Dresden, Dresden, Germany. [4]GMFC, Departamento de Física de Materiales, Universidad Complutense de Madrid, Madrid, Spain. [5]Department of Physics and Astronomy, University of Nebraska-Lincoln, Lincoln, NE, USA. [6]Integrated Quantum Optics, Institute for Photonic Quantum Systems (PhoQS), Paderborn University, Paderborn, Germany. [7]Department of Physics, University of Warwick, Coventry, UK. [8]Department of Materials, London Centre for Nanotechnology, Imperial College London, London, UK. [9]SuperSTEM, SciTech Daresbury Science and Innovation Campus, Block J, Daresbury, UK. [10]School of Chemical and Process Engineering and School of Physics and Astronomy, University of Leeds, Leeds, UK. [11]ct.qmat: Dresden-Würzburg Cluster of Excellence—EXC 2147, TU Dresden, Dresden, Germany. [12]These authors contributed equally: Andrew Rogers, Kristina Holsgrove. ✉e-mail: kholsgrove@qub.ac.uk; m.gregg@qub.ac.uk

