## [Transparent Peer Review file · Nature Communications]

Polar Discontinuities, Emergent Conductivity, and Critical Twist-Angle-Dependent Behaviour at Wafer-Bonded Ferroelectric Interfaces

Corresponding Author: Professor John Gregg

Version 0:

Reviewer comments:

Reviewer #1

(Remarks to the Author)

This manuscript aims to discuss the origin of interfacial conduction and the twisted angle dependency of interfacial conductivity as well as the domain inversion via thermocompression. The authors provide experimental data and DFT calculations to show the conductivity in H2H interface caused by the localized charge carrier rather than oxygen vacancy. However, the relaxed atomic model is not related to the experimental results of the atomic structure and there is missing a T2T atomic interface for their claim, which is crucial to identify the interfacial structural difference between H2H and T2T. As a result, it is hard to determine the authors' claims. If the relaxed structure agrees with the atomic structures of H2H and T2T, the DFT results can fully confirm the authors' claim.

In the perspective of the twisted angle dependency, this work is not yet finished as the authors mentioned in the conclusion part. That is quite weird that the authors acknowledge what they can do but not to do without any reasons. It would be much better to complete the whole study. Besides, if the domain inversions are caused the diffusion of Li2O, the domain inversions should be also observed in the H2H without twisting angle under the same thermocompression treatment. Why is it related to the twisted angle? The authors should provide the evidence to this claim.

In this manuscript, the authors provide their insights into interfacial conduction and their new findings about twisted-angle dependency domain inversion. However, some of the figures cannot fully match the results and the authors' claims. The authors need to check again the data and their claims and provide some more necessary data.

The obvious problems can be found as follows:

1. In Fig. 1c, the inset might be a bit confusing to the audience that put the the O atoms much larger than other atoms.
2. It is very difficult to notice the Nb-Nb separation by comparing the images and the line profile. As I have tried it in ImageJ, only upper part of the highlighted part is $\sim 2.2 \text{ \AA}$, the middle part and the bottom part are about $\sim 2.6 \text{ \AA}$. The authors are suggested to check again the highlighted part. Alternatively, the authors can use geometric phase analysis to show the strain or by other similar methods.
3. In Fig. S2.2 & S2.3, it is suggested to show the original spectra of the EELS.
4. Comparing the data in Fig. S2.1 & S2.3, it is quite weird that the oxygen intensity is lower in the H2H interface in S2.1, while the signals are almost the same in 3 different positions as shown in S2.3 (middle one). The line profile in Fig. S2.1 is supposed having the same integration width as box 1 in Fig. S2.3, and hence the average O intensity in Fig. S2.1 should be almost the same. Or, it is supposed to have lower integrated signal in the middle panel of Fig. S2.3. Can authors explain the difference between S2.1 and S2.3?
5. In Figure S5, it is suggested to show the modelled LNO structure along [1-100] which is the same as the Fig. 1 & 3 for better discussion. It is important to compare the theoretical and experimental data in this manuscript because the authors can directly compare the relaxed model of H2H and T2T with the atomic structures. It can also confirm the Nb-Nb separation in Fig. 1. Hence, the theoretical study in Fig. 3 will be more solid.
6. Although the cAFM signal in T2T interfaces have no contrast, the authors should also show it in Fig. S6. For example, the figure can be like Fig. 2 or Fig. S8 for better comparisons for the audience.
7. In Fig. S7, is it possible that the authors put the Fig. S7B & S7D in the opposite panels (red scale bar)?
8. The authors should provide the convergence semi-angle and the collection semi-angle (MAADF and HAADF). Also,

some more details of EELS should be also provided, for example, collection semi-angle and energy resolution etc.

Reviewer #2

(Remarks to the Author)

In manuscript, the authors construct well-bonded untwisted and twisted ferroelectric interfaces, by using high temperature thermocompression bonding of single domain. It is well-known that the charged H2H ferroelectric domain walls are extreme instable both in experiments and theoretical calculations. Therefore, the H2H walls can not be constructed by PLD or MBE epitaxial growth. High temperature thermocompression provides a new option to construct H2H domain walls. Nevertheless, the band bending at charged H2H DWs induced metallic-like conductivity is not a new phenomenon, which has been proposed in many theoretical works, such as references 34-37. The interesting phenomenon occurred at the twisted charged domain walls. The authors report that the polarization in the interface reversed its orientation, and domain wall forms alternant T2T and H2H types, rather than the initial H2H one. The T2T shows insulation characteristic. This phenomenon is very interesting and worthy of being reported. I would recommend its publication in the Nature communication after the authors address the questions below.

1. The 60° twisted interface maintains H2H polar discontinuity, and shows enhanced conductivity. It would be better if compared the electrical discrepancy between the 60° twisted (figure 4) and untwisted domain walls (figure 2). I guess the twist likely promote the conductivity compared with untwisted one.
2. Why the T2T interface shows no signs of a distinct enhanced conductivity in figure 5D? According to DFT calculation, the T2T could also induce the metallic-like conductivity.
3. The flaw in this manuscript lies in lack of underlying physical mechanism to support the phenomena that observed in twist interfaces. For example, the H2H becomes T2T domain wall, and H2H walls form kinks in Figure 5C. More detail discussions of mechanism are necessary.
4. What are the dynamic properties of charged H2H domain walls (move or maintain) under an electric field?

In addition, the author states that "Equally, interfaces in non-magnetic systems can themselves be magnetic and, in non-polar systems, ferroelectric." in the introduction. In fact, the twin wall in CaTiO₃ only exhibits spontaneous polarization due to displacements of Ti atom. But the polarization cannot be switch by electric field. According to recent work[<https://doi.org/10.1103/PhysRevLett.132.176801>], the non-polar SrTiO₃ is also a typical material that shows the twin-wall-mediated ferroelectric-like behavior.

Reviewer #3

(Remarks to the Author)

The authors present a compelling study on twist-angle-dependent phenomena at wafer-bonded ferroelectric LiNbO₃ interfaces, demonstrating emergent conductivity linked to polar discontinuities and its collapse at specific "magic" twist angles. The work extends "twistronics" beyond van der Waals materials into covalently/ionically bonded oxides using high-temperature thermocompression, revealing novel microstructural transformations and electronic behaviors.

While the study is conceptually innovative and experimentally robust, several critical aspects require further clarification and deeper analysis to fully substantiate the proposed mechanisms and broader implications. Detailed comments are provided below:

- 1) The DFT analysis (Fig. 3D-F) assumes idealized H2H and tail-to-tail (T2T) interfaces but may not account for ionic reconstruction at high bonding temperatures (~1100°C), where lithium diffusion could alter local stoichiometry. Such effects could overestimate the predicted band bending and valence state shifts at the H2H interface. To validate the semimetallic character, additional DFT calculations using hybrid functionals (e.g., HSE06) are recommended, incorporating temperature-dependent vibrational effects via ab initio molecular dynamics snapshots.
- 2) In Fig. 3D-G, the DFT model lacks twist-angle considerations, central to the study's focus. To align with the experimental emphasis, DFT calculations should be extended to include twisted interfaces at representative angles (e.g., ~21°). This would clarify how moiré patterns modulate the electronic band structure and charge carrier localization. Additionally, uncertainties in lithium positioning, which may affect the accuracy of predicted band shifts, should be addressed.
- 3) The observed inversion depth (~15 μm, Fig. 5C) is remarkably large. It remains unclear whether this depth is thermodynamically driven, representing a minimum-energy state, or kinetically limited, dependent on diffusion or bonding duration. The authors are encouraged to systematically vary bonding duration at a fixed temperature to evaluate the time dependence of the inversion depth. If kinetically limited, an Arrhenius analysis could derive activation energies, potentially linking the process to lithium diffusion or domain-wall motion.
- 4) LiNbO₃'s high Curie temperature (~1140°C) permits bonding within the ferroelectric phase, but many other oxides (e.g., BaTiO₃, PZT) have lower TC values. The authors should discuss the feasibility of applying this approach to such materials, addressing whether reduced bonding temperatures might compromise polar stability or emergent conductivity. Additionally, the generalizability of twist-angle effects beyond trigonal systems should be explored to clarify the broader applicability of the findings.
- 5) The cAFM measurements presented in Fig. 5D reveal conductivity at the H2H domain walls, but not at the T2T interface. To elucidate the structural origins of this conductivity, atomic-resolution HAADF-STEM should be employed to characterize

atomic-scale changes across H2H domain walls. Furthermore, to enhance clarity, the authors should quantify the conductivity difference (e.g., through current histograms) and discuss whether the T2T interface exhibits any residual conductivity compared to the bulk.

Version 1:

Reviewer comments:

Reviewer #1

(Remarks to the Author)

The authors have demonstrated innovative twist-angle-dependent domain inversion in thermocompression-bonded lithium niobate. The extensive efforts and the amendments are highly appreciated. This manuscript is well-suited for publication in Nature Communications. The authors may consider the following constructive suggestions to further strengthen the interpretation of their results.

The current structural evidence (e.g., Nb-column continuity in non-twisted H2H interfaces, Fig. 1, and coherent bonding at $\sim 60^\circ$ twist, Fig. 4) already provides valuable insight into interfacial strain. Nonetheless, a targeted comparison of atomic structure between H2H and T2T configurations could significantly enhance the discussion of cAFM measurement asymmetry and the mechanism of polar inversion.

Specifically, high-resolution HAADF-STEM imaging of a non-twisted T2T interface (as likely realized in the macroscopically conducting samples of Figs. S7-S8) would be highly informative. Although Li positions remain challenging to resolve, the Nb–Nb intercolumnar spacing near the interface is readily measurable and structurally meaningful. Given the DFT prediction of opposing band-bending at H2H versus T2T junctions, one might expect expanded Nb–Nb separation in T2T.

As readers, we would like to know the structural differences between H2H and T2T via thermocompression. The relationship among twist-angle, structural change, band-bending and the polarization switching could be more elaborated. Reviewer #3 is also interested in the atomic structure in the twisted sample. This might be challenging. The non-twisted T2T interface might provide insight into it.

One final question regarding the twisted samples: in Fig. S12, the cAFM shows no signal which is consistent with the previous untwisted T2T behavior. Does the inverted T2T interface in these samples still exhibit macroscopic conductivity, as seen in Fig. S8?

We emphasize that these are suggestions for improvement, not demands for infeasible tasks, such as DFT on thousands of atoms or exhaustive twist-angle mapping. A targeted T2T structural measurement would meaningfully elevate an already strong manuscript.

Reviewer #2

(Remarks to the Author)

I thank the authors for their thorough and thoughtful responses to my comments. The revisions and additional analyses have significantly strengthened the manuscript. Most of my initial concerns have been adequately addressed. While a more detailed underlying physical mechanism, especially regarding the exact electronic structure modulation at twisted interfaces, remains somewhat underexplored, the phenomena reported here are highly interesting and represent an important step forward in the field of interface engineering in ferroelectrics. Given the novelty, clarity, and potential impact of the findings, I believe the manuscript in its current form is worthy of publication in Nature Communications.

Reviewer #3

(Remarks to the Author)

I have reviewed the revised version of your manuscript and your detailed response to the reviewers' comments. I am very pleased with the revisions you have made. I have no further queries and support the acceptance of your work without any modifications.

Version 2:

Reviewer comments:

Reviewer #1

(Remarks to the Author)

I would like to thank the authors for their detailed and thoughtful responses to my comments. I appreciate the additional efforts made to characterize the T2T interfaces using HRTEM with GPA, despite the limited availability of SuperSTEM time. Most of my concerns have been adequately addressed. Although the structural information is still under exploration, which is one of the keys to elucidating a more detailed physical mechanism, the findings of interfacial ferroelectric engineering demonstrate high novelty. I have no further suggestions and support the acceptance of the revised manuscript.

Response to Reviews on NCOMMS-25-57898-T: “Polar Discontinuities, Emergent Conductivity, and Critical Twist-Angle-Dependent Behaviour at Wafer-Bonded Ferroelectric Interfaces” by Rogers *et al.*

Reviewer 1:

Dealing initially with the specific list of comments raised by reviewer #1:

Comment 1: “In Fig. 1c, the inset might be a bit confusing to the audience that put the O atoms much larger than other atoms.”

Response 1: In the original draft for this inset, we used a representation of the crystal structure of LiNbO₃ (LNO) in which the sizes of the chemical species were reflective of their ionic radii (oxygen has an ionic radius around twice that of Nb⁵⁺ and Li⁺)¹. This is often the approach used to represent structures. However, we accept that, as rendered, the inset did not allow the reader to easily visually compare the distribution of Nb ionic columns, in the idealised LiNbO₃ structure, with that in the HAADF STEM image. We have therefore changed the inset in several ways, such that: the “ball and stick” representation is now used for multiple unit cells; in the lower part of the inset, the oxygen and lithium are removed to reveal Nb only. We think this now allows for an easy direct comparison between the expectation of Nb column distribution and the experimental observation.

Fig. R1: Scanning transmission electron microscopy (STEM) analysis of bonded LNO interface. (A) Schematic of bonded H2H sample. (B) DPC-STEM on bonded interface region; the center of the interface is marked with grey dashed lines. (C) HAADF-STEM of the bonded interface displaying niobium atomic columns viewed along $[1\bar{1}00]$; a schematic of the relative positions of the different chemical species in the LNO structure, as viewed along $[1\bar{1}00]$, is shown by the “ball and stick” representation in the upper part of the inset. The lower part of the inset is the same projection in which only the Nb atoms are rendered (below). This can be directly compared with the HAADF image. (D) Higher magnification HAADF-STEM image from the red boxed region in (C). (E) Fast Fourier Transform (FFT) of D showing apertures moved through the 0006 Bragg peaks. (F)-(G) Inverse FFTs performed for the green, and orange apertures in (E). (H) Nb-Nb spacing measured along the c-axis within (D); positive and negative directions refer to Nb-Nb pairs in the top crystal and bottom crystal respectively, moving away from the interface. Shading indicates calculated standard deviation obtained by averaging across multiple atomic rows.

Changes to the Manuscript: We have changed the inset in Fig.1c (see figure R1 above) to use representations of the crystal structure which allow a much more obvious comparison between the ideal Nb ionic column distribution and that observed experimentally.

Comment 2: *“It is very difficult to notice the Nb-Nb separation by comparing the images and the line profile. As I have tried it in ImageJ, only upper part of the highlighted part is ~2.2 Å, the middle part and the bottom part are about ~2.6 Å. The authors are suggested to check again the highlighted part. Alternatively, the authors can use geometric phase analysis to show the strain or by other similar methods.”*

Response 2: We agree that the changes in the Nb-Nb separations are subtle and difficult to see without reasonably extensive analysis. In our original manuscript, panel H in Fig. 1 presented the average of all Nb-Nb separations between each row of ionic columns occurring sequentially along the +c (in one crystal) and -c (in the other crystal) directions; these separations were then presented as a function of position, relative to the perpendicular “distance” from the bonded interface. In detail, the Nb-Nb separations were determined by fitting Gaussians to line profiles and finding the average separations between Gaussian peaks for each Nb row.

We note that this was not the only methodology used and described in the original manuscript. As presented in Fig. 1E-G, we also used digital apertures to select specific regions of intensity along the reciprocal crystallographic $\pm c^*$ directions of the LiNbO_3 reciprocal lattice, as revealed in the Fast Fourier Transform (FFT) of the real space image. Inverse FFTs, using apertures further from the origin in reciprocal space (and hence associated with smaller distances in real space), produced images in real space of Nb columns in the interfacial region; on the other hand, inverse FFTs using apertures closer to the origin in reciprocal space (and hence associated with larger distances in real space) revealed real space intensity from Nb columns further from the interface. This digital dark-field imaging thus confirmed our conclusions from direct real-space measurements described above, that the Nb-Nb separations were smaller in the interfacial region than elsewhere.

In addition to these two approaches, we have now performed two further methods of image analysis on the untwisted H2H system: geometric phase analysis (as suggested by the reviewer) and the determination of interatomic separations between atomic positions using Atomap (a python package for identifying atomic columns and measuring interatomic distances). The results are illustrated below in figure R2. As can be seen, both techniques confirm the compression of the Nb-Nb separations along directions parallel to the polar axes, at the bonded interface.

Changes to the Manuscript: We believe that the additional analyses requested by the reviewer will be useful to readers and have included them in the supplementary information associated with the manuscript, as a new Fig. S2.

Figure R2: Two additional techniques have been used to ensure that the claimed reduction in Nb-Nb interatomic separation along the [0001] direction at the interface is robust. In (A) Atomap (a python package for identifying atomic columns and measuring distances between them – see <https://atomap.org/>) has been used to identify the Nb column positions and variations in interatomic positions – note the distinct dark band at the interface in which relative separations are distinctly reduced. Here the scale bar indicates the colours associated with different fractional variations of interatomic separations relative to a reference (representative of bulk). In (B), conventional Geometric Phase Analysis (GPA) has been used. This image shows ε_{yy} where y is the direction parallel and antiparallel to the polarisation directions in the two crystals (marked by white arrows). Again, distinct compression in interatomic separations along the [0001] can be seen. Here, the scale bar shows colours associated with differing values for the ε_{yy} components in the 2D strain tensor, calculated with respect to a unit mesh from the bulk region of the LNO.

Comment 3: “In Fig. S2.2 & S2.3, it is suggested to show the original spectra of the EELS.”

Response 3: The spectra given in these figures are the original EEL spectra.

Changes to the Manuscript: None.

Comment 4: “Comparing the data in Fig. S2.1 & S2.3, it is quite weird that the oxygen intensity is lower in the H2H interface in S2.1, while the signals are almost the same in 3 different positions as shown in S2.3 (middle one). The line profile in Fig. S2.1 is supposed having the same integration width as box 1 in Fig. S2.3, and hence the average O intensity in Fig. S2.1 should be almost the same. Or, it is supposed to have lower integrated signal in the middle panel of Fig. S2.3. Can authors explain the difference between S2.1 and S2.3?”

Response 4: We are somewhat at a loss in responding to this comment. To help in the discussion, we have copied Fig. S2.1 below. Our interpretation of the data in this figure (represented by the intensity-position information extracted and represented in the right-hand panel) is that indeed, within error, there is no evidence of a reduced oxygen intensity at the interface. Perhaps the reviewer is referring to the darker contrast bands in the EELS colour maps. These dark regions develop from carbon build-up and are not evident in the extracted profiles in the right-hand panel. We hence do not see the oxygen intensity dip, outside of uncertainties, to which the reviewer refers. We therefore think that there is indeed consistency between Fig. S2.1 and Fig.S2.3, as expected by the reviewer. The take-home message from this analysis is still therefore that oxygen stoichiometry seems to be maintained at the interface, within error.

We emphasise, however, that even a slight oxygen non-stoichiometry (below our error levels), is still unlikely as the primary origin for the emergent conductivity at the charged

H2H interfaces for two reasons: firstly, our experiments involving the deposition and oxidation of thin aluminium films that were designed to induce significant deoxidation of the LiNbO_3 , did not lead to enhanced conductivity (illustrated in Fig. S5 and discussed in both the main manuscript and supplementary information); secondly, the existence of a polarisation discontinuity in naturally formed H2H domain walls in LiNbO_3 is consistently seen in other work to be sufficient to generate emergent conductivity through band-bending of the electronic potential landscape. Emergent interfacial conduction is therefore fully expected, irrespective of local oxygen stoichiometry.

Changes to the Manuscript: None.

Figure S2.1. MAADF-STEM and EELS data for untwisted H2H sample. Line profile averaged across entire O-K map and Nb $M_{4,5}$ map (in direction of arrow) reveals homogeneous (within error) concentration of O and Nb transitioning from one crystal (across interface) to the other crystal. Area corresponds to Figure 3 in main paper.

Comment 5: “In Figure S5, it is suggested to show the modelled LNO structure along $[1-100]$ which is the same as the Fig. 1 & 3 for better discussion. It is important to compare the theoretical and experimental data in this manuscript because the authors can directly compare the relaxed model of H2H and T2T with the atomic structures. It can also confirm the Nb-Nb separation in Fig. 1. Hence, the theoretical study in Fig. 3 will be more solid.”

Response 5: We agree with the reviewer that a rotation of the LiNbO_3 structure in the supplementary figure is sensible.

Changes to the Manuscript: The modelled LNO structure has been rotated such that it is now viewed along the $[1-100]$ direction, to be consistent with Figs. 1 & 3 in the main manuscript. Note this figure is now Fig. S6 in the resubmitted manuscript.

Comment 6: “Although the cAFM signal in T2T interfaces have no contrast, the authors should also show it in Fig. S6. For example, the figure can be like Fig. 2 or Fig. S8 for better comparisons for the audience.”

Response 6: This is a sensible comment too, and we agree that this cAFM map for the T2T bonded interfaces should be included.

Changes to the Manuscript: The new Fig S7 has been modified to include a panel (new panel “B”) with cAFM data, making things more consistent with Fig. 2 and Fig. S8. See below:

Comment 7: “In Fig. S7, is it possible that the authors put the Fig. S7B & S7D in the opposite panels (red scale bar)?”

Response 7: The reviewer is correct, and we apologise for the mistake.

Changes to the Manuscript: We have corrected the supplementary figure (now Fig. S8 in the revised manuscript) to that given below:

Comment 8: “The authors should provide the convergence semi-angle and the collection semi-angle (MAADF and HAADF). Also, some more details of EELS should be also provided, for example, collection semi-angle and energy resolution etc.”

Response 8: We agree with the reviewer and apologise for this omission.

Changes to the Manuscript: The following additional information has been added to the supplementary information in the Materials and Methods section:

“Atomic-resolution STEM images were acquired on a Nion UltraSTEM 100 operated at 100kV, with a convergence semi-angle of 30mrad and a probe current of ~30pA. The collection semi-angle range for HAADF-STEM images was 89-195 mrad and MAADF-STEM images was 47-85 mrad. Electron Energy Loss Spectroscopy (EELS) was acquired on a Gatan UHV Enfina EEL spectrometer retrofitted with a Quantum Detectors Merlin EELS hybrid-pixel detector. The EELS collection semi-angle was 35 mrad and the effective energy resolution was 0.4eV.”

More General Comments Made by Reviewer #1 (discussed as “A”, “B” and “C”):

- A.** *“The authors provide experimental data and DFT calculations to show the conductivity in H2H interface caused by the localized charge carrier rather than oxygen vacancy. However, the relaxed atomic model is not related to the experimental results of the atomic structure and there is missing a T2T atomic interface for their claim, which is crucial to identify the interfacial structural difference between H2H and T2T. As a result, it is hard to determine the authors’ claims. If the relaxed structure agrees with the atomic structures of H2H and T2T, the DFT results can fully confirm the authors’ claim.”*

Response: The reviewer is bringing up a good point, concerning the extent to which we have been able to correlate atomic positional information between experiment and theory. We have tried extensively to do this, but have run into several difficulties which are currently insurmountable:

- (i)** Modelling image contrast (using Dr. Probe [Ultramicroscopy **193**, 1-11 (2018)]) for both bright and dark field STEM images and comparing models with our data to find out where atomic columns reside, shows that we cannot accurately locate lithium. Image modelling shows that lithium could be located anywhere within the unit cell without any measurable difference expected in either the dark-field or bright-field images. Niobium columns are well resolved, using HAADF and MAADF detectors and oxygen columns can be reasonably located using ABF and BF imaging (see figure R3 below). However, full experimental ionic location (including lithium) is not possible with our current imaging capabilities (even in SuperSTEM).
- (ii)** In our experiments, it is clear that the Nb-Nb interionic separation, measured along a vector perpendicular to the bonded interface, decreases in the interfacial region (see “Response 2” above). Further imaging and analysis have revealed that this directional unit cell compression is accompanied by an associated expansion parallel to the interface (as might be expected through Poisson ratio-type arguments). To accommodate this expansion along the bonded interface, it undergoes a physical corrugation, with a period of ~8nm in one dimension (see figure R4). 8 nm is on the order of 15 unit cell lengths, if measured along one of the in-plane primary lattice parameter directions. With similar behaviour in symmetry-related directions on the (0001) interfacial plane, this would mean that hundreds of unit cells and thousands of atoms would be needed to capture the existence of this physical modulation and allow DFT modelling to calculate the relaxed atomic structure in an experimentally realistic context. Such large atomic populations are currently out-of-the-scope of DFT.

Figure R3: Simulations performed using Dr. Probe utilising the STEM detector collection angles experimentally used at SuperSTEM. The LiNbO_3 unit cell for each set of simulations is shown on the left (viewed down $[1-10]$ zone axis). It is apparent that altering the specimen thickness from 10nm to 30nm (more comparable to the real TEM specimen) has little change on the STEM images produced: the HAADF and MAADF images display the heaviest atom, Nb, while the ABF and BF images display both the heaviest atom, Nb, and the next heaviest atom, O. The Li atoms are not observed in any detector image. However, when the input unit cell was altered so that the heaviest atoms, Nb, were artificially deleted (bottom row of images) then there was a faint trace of the lightest atoms (Li) in the resultant ABF and BF images. We can conclude from these simulations that when Nb is present (which is reflective of the genuine unit cell of LiNbO_3) it is not possible to visualise nor identify the exact location of Li ion columns.

Figure R4: DPC contrast of a lamella cut perpendicular to a bonded interface, supporting a H2H polarisation discontinuity, reveals the rippled nature of the interface morphology. To facilitate this rippling, the total interface area must have increased, and this is consistent with observations made of unit cell compressions in directions parallel to the $[0001]$ polar axes and associated unit cell expansions needed parallel to the bonded interface itself. The ripple period is of the order of 8nm. The colour wheel indicates the polar directions in the image.

Notwithstanding the issues discussed above, we think that it is very important to note that the DFT predictions and experimental observations of functional (as opposed to structural) properties at charged interfaces can be meaningfully compared: DFT predicts emergent conductivity in a tightly located region at the bonded H2H interface and this is confirmed using cAFM and mesoscale two probe current-voltage measurement. DFT also predicts emergent conductivity at the bonded T2T interface, but the conduction expected is not nearly so spatially confined. This is again confirmed by experiment, where T2T interfaces do show mesoscale two probe conduction well above bulk levels, but no clearly resolved cAFM signal. It therefore seems that the accuracy of functional expectations arising from DFT are robust to the structural uncertainties discussed above. We therefore do not agree with the statement made by the reviewer “*As a result, it is hard to determine the authors’ claims*”, because the thrust of the manuscript is not primarily concerned with atomic structure; rather, our key message involves the effects of interfacial twist on emergent conductivity and screening properties at the bonded interfaces. Conductivity and associated screening are expected from DFT modelling of untwisted bonded LiNbO₃ and this expectation is experimentally confirmed.

Changes to the Manuscript: We have added the new STEM image simulations given in figure R3 to the supplementary information to show how Nb and O can be reasonably experimentally located, but Li cannot.

B. “*In the perspective of the twisted angle dependency, this work is not yet finished as the authors mentioned in the conclusion part. That is quite weird that the authors acknowledge what they can do but not to do without any reasons. It would be much better to complete the whole study.*”

Response: We made the comment in our initial draft: “*Of course, we acknowledge that a great deal more needs to be done*” as a recognition of the huge phase space associated with twisting. After all, superconductivity in twisted graphene can be realised or destroyed by changing the twist angle by as little as $\sim 0.1^\circ$. A full exploration of all twists in bonded LNO samples between 0° and 60° , with an angular resolution below 0.1° , would imply making and fully characterising something of the order of 1,000 samples. This is a mammoth task, which would take many years to complete. While a full phase space exploration may show more, the samples that we have made to date nevertheless reveal interesting and important twist-related physics that we think is worthy of report.

We reiterate that our study is the first in which twist dependence of the emergent functional (transport) properties of charged ferroelectric interfaces has been explored. It is the first to report specific twist angles at which electrical screening appears to be inhibited, resulting in uncompensated bound charge and electric fields sufficient to cause local polar switching (domain inversion). It is also the first to link the inability for screening charges to compensate bound charge to the specific angles at which the coincident lattice parameters diverge (become sparse) and local aperiodicity arises.

Changes to the Manuscript: as a result of this reviewer comment, we have become concerned that important aspects of novelty have not been made clear enough in our initially submitted draft and so have made significant effort to alter the main text to make the key thrust of the paper more transparent. The significant changes can be seen in the marked up version of the resubmitted manuscript (changes in red).

C. “*Besides, if the domain inversions are caused the diffusion of Li₂O, the domain inversions should be also observed in the H2H without twisting angle under the same thermocompression treatment. Why is it related to the twisted angle? The authors should provide the evidence to this claim.*”

Response: Again, we suspect that we have not communicated the origin of the domain inversions in our study carefully enough. In short, we do not think the domain inversion in our work has anything to do with enhanced Li₂O diffusion out of the system at specific twist angles. Rather we believe that the aperiodicity associated with divergence in the coincident lattice at specific twist angles alters the electronic properties at the interface

such that screening of the positive bound charge at the H2H interfaces becomes impossible. The unscreened positive bound charge generates a depolarising field large enough to overcome the coercive field of the LNO (perhaps only at elevated temperature), causing domain inversion. The discussions on prior work on inversions in bulk single crystals (where Li_2O has been suspected as being key) were meant to illustrate that the inversion phenomenon can happen when surface-related electric fields are generated and when effective screening cannot be realised. We have tried to emphasise this more clearly in the resubmitted version of the manuscript.

For completeness, in response to this comment and a comment from reviewer #3, we have also measured lithium concentration profiles as a function of position from the bonded interfaces in samples with and without domain inversion. We see no lithium concentration variation within error in either sample (see response to Comment 1 by Reviewer #3 below) showing that the dipolar inversion results from unscreened fields that are unrelated to lithium diffusion.

Reviewer 2:

Dealing initially with the list of specific comments raised by reviewer #2:

Comment 1: *“The 60° twisted interface maintains H2H polar discontinuity and shows enhanced conductivity. It would be better if compared the electrical discrepancy between the 60° twisted (figure 4) and the untwisted domain walls (figure 2). I guess the twist likely promote the conductivity compared with untwisted one.”*

Response 1: We appreciate the reviewer’s intuition on this point. The cAFM currents presented on the H2H samples without deliberate twist definitely look to be less impressive than those seen in the H2H with $\sim 60^\circ$ twist. We do not think, however, that a higher conductivity for the 60° sample can be robustly concluded. Our opinion is based on several factors:

- (i) firstly, it is clear that the cross-section of the untwisted H2H sample has polishing scratches, while the 60° H2H does not. We think that the spotted cAFM contrast in the untwisted sample results from these scratches interrupting good sample-tip contact, while the smooth polish on the $\sim 60^\circ$ twisted sample allows for much better cAFM contact at all points along the bonded interface;
- (ii) secondly, both the mesoscale measurements, made using GaInSn liquid electrodes and microprobes, and the cAFM are two-probe in nature. This means that currents will depend on the contact resistance, as well as the inherent resistance of the 2DEG at the bonded interface. As the contact resistance may vary considerably, we are reticent to conclude much from direct current comparisons.
- (iii) thirdly, we note that in mesoscale two probe measurements, the 60° and untwisted samples gave similar I-V characteristics, with similar current levels. Notwithstanding (ii) above, we think this points to modest, if any, differences in interfacial conductivity.

We accept that a detailed set of 4-probe measurements would be desirable on interfaces as a function of twist angle. However, our attempts to do this have been foiled by contact issues and time-dependent changes in conductivity, which we do not yet understand. Overcoming this frustrating phenomenon, and understanding its origin, are objectives for future work, but success may be on the timescale of years. Moreover, any subtle changes in interfacial conductivity are not the thrust of the message communicated in this manuscript. Instead, we focus on the very obvious differences between the screening capabilities of H2H interfaces at most twist angles with those at particular angles of twist ($\sim 21^\circ$, $\sim 14^\circ$, $\sim 74^\circ$) where the size of the coincident lattice tends to infinity and aperiodicity is introduced.

Changes to the Manuscript: None.

Comment 2: “Why the T2T interface shows no signs of a distinct enhanced conductivity in figure 5D? According to DFT calculation, the T2T could also induce the metallic-like conductivity.”

Response 2: We agree that the observations, such as in Fig. 5D, are somewhat unexpected. We had thought that cAFM signals at the T2T bonded interfaces should appear. However, we have not seen measurably enhanced cAFM profiles along the bond-line itself in any T2T samples examined to date. It must, however, be noted that, when we perform mesoscale I-V characterisation, we do see enhanced conductivities along T2T bonds that are noticeably higher than bulk LiNbO₃ (Fig. S8 C&D). It should also be noted that from literature, the same conflict also appears to be true in regular domain wall research. While several studies report enhanced conductivity at both H2H and T2T domain walls in LiNbO₃ by mesoscopic transport measurements [*Physical Review Applied* **17**, 4, (2022): 044011], an enhanced conductive AFM contrast only occurs at head-to-head walls, insofar as we are presently aware [*Journal of Applied Physics* 132.4 (2022); *Physical Review Applied* 10.3 (2018): 034002]. There are several potential reasons why this might be the case. Firstly, pertaining to our own modelling work, the DFT information shows that the atomic and the electronic structure of the bonded H2H and T2T interfaces are fundamentally different, resulting in a more diffuse spatial distribution of conduction in T2T as compared with H2H interfaces (Fig. 3G is an illustration of this). The density of states at the Fermi level, as calculated for the ideal structures at H2H and T2T interfaces, is also different. While the localisation for T2T walls is still below the typical pixel resolution of a c-AFM scan, the difference in confinement could perhaps change the peak conductivity that the interface exhibits, which may manifest as a lower measured cAFM current.

Changes to the Manuscript: We have added a panel to Fig. S7 (panel B) to make it clear that cAFM does not pick up conduction at T2T bonded interfaces. Fig. S8 also shows that this absence of cAFM goes hand-in-hand with reasonably significant conduction detected using the mesoscale liquid electrode two-probe technique, illustrating that enhanced conductance is present, but for a host of potential reasons (including differences in conductivity, carrier type and spatial confinement) is not detectable in cAFM.

Comment 3: “The flaw in this manuscript lies in lack of underlying physical mechanism to support the phenomena that observed in twist interfaces. For example, the H2H becomes T2T domain wall, and H2H walls form kinks in Figure 5C. More detail discussions of mechanism are necessary.”

Response 3: We should here emphasise the key messages that we had hoped to convey in the manuscript:

- (i) that bonded interfaces in which H2H polar discontinuities are maintained, show highly localised interfacial conductivity, consistent with a 2DEG;
- (ii) that the successful creation of H2H discontinuities and associated conductivity are generally insensitive to twist;
- (iii) that for some very specific twist angles ($\sim 14^\circ$, $\sim 21^\circ$ and $\sim 74^\circ$), local dipolar inversion occurs adjacent to the bonded interface – this phenomenon is not seen at any of the other twist angles investigated;
- (iv) that the reversal of the polar orientation (local switching) must be due to a positively charged electric field source at the interface large enough to overcome the coercive field for LNO;
- (v) the obvious source of interfacial positive charge is the bound charge associated with the H2H polar discontinuity and hence observations suggest that, at specific twist angles, the electronic behaviour of the interface becomes such that complete screening of bound charge cannot be realised;
- (vi) the angles at which ineffective screening occurs ($\sim 14^\circ$, $\sim 21^\circ$ and $\sim 74^\circ$) are special, insofar as they coincide with the non-intuitive divergences in the coincident lattice periodicity for hexagonal mesh lattice point distributions;

- (vii) we therefore conclude that it is twist-induced aperiodicity that leads to a fundamental change in electronic behaviour (the emergence of pseudo-band gaps) and the inability of the interface to facilitate screening of the bound charge;
- (viii) the localised domain inversion and polar switching is spatially limited, because the naturally forming H2H domain walls that exist within both single crystals are strongly conducting and can screen electric fields.

We accept, however, that the points listed above should have been made more explicitly in the original manuscript and so we have made significant changes to the text to emphasise that we are indeed putting forward a sensible and interesting fundamental mechanism for the twist-related polar inversion seen. We note that, in literature, the kinks / corrugations seen in naturally forming domain walls are due to imperfect charge screening [Marton, P. *et al.* Zigzag charged domain walls in ferroelectric PbTiO₃. *Phys. Rev. B* **107**, 094102 (2023)]. However, we also note that the very shallow angles of corrugations we observe in the H2H domain walls (as opposed to H2H bonded interfaces) suggest that the screening is almost perfect, commensurate with the strong domain wall conductivity seen in cAFM scans.

Changes to the Manuscript: We have made extensive changes to the text of the manuscript to be much more explicit about the key features in the observations and the proposed underlying mechanism involved, as presented in points (i)-(viii) above. We have also added a note concerning the kinks in the H2H domain walls that form as a result of local polar inversion, along with the reference noted above.

Comment 4: “What are the dynamic properties of charged H2H domain walls (move or maintain) under an electric field?”

Response 4: All the conducting H2H walls (including the H2H interfacial polar discontinuity) appear to be mobile under applied electric fields. In a number of new experiments, we have deposited thin film coplanar electrodes, with interelectrode gaps in which either the conventional H2H domain walls, or the H2H bonded interface exist. Bias was then applied and the spatial position of the polar discontinuity mapped using in-situ PFM. As can be seen in Figs. R5-R7 below, both domain walls and charged interfaces are mobile under bias.

Changes to the Manuscript: While these observations are certainly interesting, we do not think that field-dependent mobility of the interfaces (natural domain walls and bonded interfaces) is central to the thrust of the manuscript and are therefore only including these results in the “response to reviewer” documentation.

Figure R5: In-situ PFM revealing H2H domain wall dynamics under applied bias. Data here comes from the conventional H2H domain walls that form after the local polar inversion induced by an unscreened bonded interface; in this instance, these H2H domain walls are in the sample twisted by $\sim 21^\circ$. (a) AFM topography with slow scan axis off. (b) PFM phase data associated with different applied bias levels (voltage levels indicated in the panel). (c) Wall location as indicated by PFM phase change across the H2H interface and the accompanying PFM amplitude dip (not shown). Scale bar is 1 μm .

Figure R6: In-situ PFM revealing H2H interface dynamics under applied bias. Data here comes from the bonded interface in a H2H sample that had no deliberate twist. (a) AFM topography from line scans with the slow scan axis off. (b) PFM phase data from line scans across the interface when under different bias levels. (c) Associated PFM amplitude. (d) Wall location (as a function of time and bias) indicated by the PFM phase change and PFM amplitude dip. (e-h) The same as above for applied voltage in the opposite sense. Scale bar is $0.3\mu\text{m}$.

Figure R7: In-situ PFM data revealing dynamics of the polar discontinuity in a twisted T2T bonded interface under bias. These data are associated with field-induced motion of the T2T polar discontinuity found after polar inversion at the bond line in $\sim 21^\circ$ twisted samples. (a) AFM topography with slow scan axis off. (b) PFM phase data as a function of different levels of applied bias. (c) Wall location as indicated by the point at which PFM phase changed and PFM amplitude dipped. Scale bar is $0.5\mu\text{m}$.

More General Comments Made by Reviewer #2 (discussed as “A” and “B”):

A *“In the manuscript, the authors construct well-bonded untwisted and twisted ferroelectric interfaces, by using high temperature thermocompressional bonding of single domain. It is well-known that the charged H2H ferroelectric domain walls are extreme instable both in experiments and theoretical calculations. Therefore, the H2H walls cannot be constructed by PLD or MBE epitaxial growth. High temperature thermocompression provides a new option to construct H2H domain walls. Nevertheless, the band bending at charged H2H DWs induced metallic-like conductivity is not a new phenomenon, which has been proposed in many theoretical works, such as references*

34-37. *The interesting phenomenon occurred at the twisted charged domain walls. The authors report that the polarization in the interface reversed its orientation, and domain wall forms alternant T2T and H2H types, rather than the initial H2H one. The T2T shows insulation characteristic. This phenomenon is very interesting and worthy of being reported. I would recommend its publication in Nature Communications after the authors address the questions below.*

Response: We agree with the reviewer and would only like to point out that all previous reports of conductivity arising at charged interfaces in LiNbO₃ (and indeed ferroelectrics generally) refer to naturally occurring domain walls. These fundamentally differ from bonded structures, especially when twist is introduced. To our knowledge, the submitted work represents the first attempt to explore the possibility of enhanced conductivity at bonded interfaces rather than naturally occurring domain walls.

Changes to the Manuscript: None.

B *“In addition, the author states that “Equally, interfaces in non-magnetic systems can themselves be magnetic and, in non-polar systems, ferrielectric.” in the introduction. In fact, the twin wall in CaTiO₃ only exhibits spontaneous polarization due to displacements of Ti atom. But the polarization cannot be switch by electric field. According to recent work[<https://doi.org/10.1103/PhysRevLett.132.176801>], the non-polar SrTiO₃ is also a typical material that shows the twin-wall-mediated ferroelectric-like behaviour.”*

Response: We agree with the summary on the CaTiO₃ domain walls presented by the reviewer and the lack of polarisation switching observed. We would emphasise, however, that this is why we refer to these walls as “ferrielectric” as opposed to “ferroelectric”. While we might debate definitions of “ferrielectric”, we have taken our lead from the use of the term in the title of the paper to which we make reference: “*Direct Observation of Ferrielectricity at Ferroelastic Domain Boundaries in CaTiO₃ by Electron Microscopy*”.

We acknowledge the additional reference given by the reviewer and the possibility of flexoferroelectricity.

Changes to the Manuscript: In the resubmitted manuscript, we have made reference to the PRL paper highlighted by the reviewer and to the possibility of flexoferroelectricity in SrTiO₃.

Reviewer 3:

Again, dealing with the specific comments raised by reviewer #3:

Comment 1: *“The DFT analysis (Fig. 3D-F) assumes idealized H2H and tail-to-tail (T2T) interfaces but may not account for ionic reconstruction at high bonding temperatures (~1100°C), where lithium diffusion could alter local stoichiometry. Such effects could overestimate the predicted band bending and valence state shifts at the H2H interface. To validate the semimetallic character, additional DFT calculations using hybrid functionals (e.g., HSE06) are recommended, incorporating temperature-dependent vibrational effects via ab initio molecular dynamics snapshots.”*

Response 1: This is an interesting point. It prompted us to experimentally map the lithium profile near to the bonded interfaces to see if any evidence for lithium out-diffusion might be seen. We used Secondary Ion Mass Spectrometry (SIMS) on a Focused Ion Beam Microscope. The results (figure R8) show that there is no significant variation of Li⁺ across a sample displaying the domain inversion or in one not displaying the domain inversion, when compared to a reference single crystal LiNbO₃ sample which was not subjected to heat treatment. This suggests that out-diffusion of lithium along the bonded interface does not occur in our work, even at the elevated temperatures associated with bonding.

Figure R8: Secondary ion mass spectroscopy imaging of the bonded head-to-head LiNbO_3 interfaces: Spatially resolved maps of the normalised, relative Li^+ concentrations in the head-to-head bonded samples that maintained their H2H configuration (A) and in those in which a local polar inversion occurred adjacent to the interface to change it to a T2T configuration (B). The black dotted line indicates the approximate location of the bonded interface, which runs horizontally. (C) Li^+ concentration profiles, comprised of multiple $50\mu\text{m} \times 50\mu\text{m}$ SIMS images taken on a trace running perpendicular to the interface, for both the non-inverted and inverted samples. The variation of Li^+ across a reference LiNbO_3 single crystal, which was not subjected to heat treatment, is shown by the green shaded area.

In our DFT, we would emphasise that the primary goal has been to explore the possibility of band bending in the bonded structures, to the extent that the interface might become semimetallic. We would note that the temperature-dependent vibrational effects expected are known to reduce the bandgap. In a previous work², some of the authors have shown, by ab initio molecular dynamics and Allen–Heine–Cardona theory, that temperature effects reduce the fundamental bandgap of LiNbO_3 by about 1 eV at 400 °C. Therefore, thermal effects will not affect the semimetallic nature of the interfaces. Concerning the effect of hybrid potentials in the electronic structure, we have performed the additional HSE06 calculations suggested by the reviewers. In agreement with the findings for the charged domain walls in bulk LNO³, hybrid DFT slightly modifies both the electronic structure and the density of state of the systems, but does not affect the predicted semimetallic nature of the interface. This prediction of the semimetallic character of the investigated systems is therefore robust, as predicted by DFT with (semi-)local XC-functionals.

Changes to the Manuscript: We have added the HSE06 calculations and the experimental SIMS mapping results to the supplementary information.

Comment 2: “In Fig. 3D–G, the DFT model lacks twist-angle considerations, central to the study’s focus. To align with the experimental emphasis, DFT calculations should be extended to include twisted interfaces at representative angles (e.g., $\sim 21^\circ$). This would clarify how moiré patterns modulate the electronic band structure and charge carrier localization. Additionally, uncertainties in lithium positioning, which may affect the accuracy of predicted band shifts, should be addressed.”

Response 2: The supercells employed to model the untwisted interface are entirely suitable for calculations within DFT or hybrid-DFT. Modelling different twist angles requires much larger supercells to preserve the periodicity of the semi-infinite bonded systems. The number of atoms (ν) which must be considered in the DFT cell can be calculated as a function of the twist angle (θ) following reference⁴. Applying the described formalism to the LiNbO₃ lattice results in Figure R9:

Figure R9: The number of atoms (ν) associated with the supercell needed for DFT modelling, as a function of twist angle (θ). Clearly, as the magnitude of the coincident lattice produced at the twisted interface changes, so too must the magnitude of the supercell required for DFT. For some of the most interesting angles of twist (e.g. $\sim 21^\circ$), where experiments show the interface to be unscreened, divergence in the magnitude of the separation between coincident lattice points occurs, making the supercell size untenable for DFT.

At angles such as $\sim 21^\circ$, the required number of atoms rapidly grows beyond many thousands, leading to supercells which are beyond the capabilities of modern supercomputers. The smallest possible cells, modelling in-plane rotations between 0° and 60° , contain about 2000 atoms. This renders the elaborate studies with twist (including in-plane and out of plane translations) de facto impossible. An exception is given for an in-plane rotation of 60° . Such a study is possible and has been performed according to the referee’s suggestion. The additional calculations, which are included in the revised version of the supplementary information, reveal little difference between 0° and 60° structures. Indeed, due to the trigonal crystal symmetry, such a twist relationship is not expected to deeply modify the atomic and electronic structure of the interface with respect to the 0° configuration. Investigation of larger structures with different twist angles might be the topic of upcoming investigations to be performed with more efficient computational schemes (tight-binding-based models for example).

Changes to the Manuscript: Calculations with 60° twisted structures have now been included in the supplementary information (Fig. S6).

Comment 3: “The observed inversion depth ($\sim 15 \mu\text{m}$, Fig. 5C) is remarkably large. It remains unclear whether this depth is thermodynamically driven, representing a minimum-energy state, or kinetically limited, dependent on diffusion or bonding duration. The authors are encouraged to systematically vary bonding duration at a fixed temperature to evaluate the time dependence of the inversion depth. If kinetically limited, an Arrhenius analysis could

derive activation energies, potentially linking the process to lithium diffusion or domain-wall motion.”

Response 3: We welcome the suggestion from the reviewer concerning a study to probe the thermodynamics and kinetics of the inversion of the polarisation at the initially H2H bonded interfaces. Indeed, this is something that we had hoped to do. However, we found that such a study is extremely non-trivial. This is mainly because creating samples with the exact twist angles needed to induce polar inversion is difficult to achieve. To give context, after around 3 years of research on this topic, with a large team of researchers, we have only produced 4 samples in which the local polar inversion phenomenon has occurred, all at angles at which the coincident lattice parameters at the interface diverge: two at $\sim 14^\circ$, one at $\sim 21^\circ$ and one at $\sim 74^\circ$. With this level of rarity, the kind of study suggested by the reviewer would take decades to complete, even if all of our efforts were focused on this.

Changes to the Manuscript: We have emphasised in the revised manuscript that the inversion phenomenon is rare.

Comment 4: *“LiNbO₃'s high Curie temperature ($\sim 1140^\circ\text{C}$) permits bonding within the ferroelectric phase, but many other oxides (e.g., BaTiO₃, PZT) have lower TC values. The authors should discuss the feasibility of applying this approach to such materials, addressing whether reduced bonding temperatures might compromise polar stability or emergent conductivity. Additionally, the generalizability of twist-angle effects beyond trigonal systems should be explored to clarify the broader applicability of the findings.”*

Response 4: The high Curie Temperature, the large ratio between the Curie Temperature and the melting temperature and the uniaxial nature of the ferroelectricity were all extremely important parameters for the selection of LNO for our research. It is the combination of these material features that allowed the sintering of the bond entirely within the ferroelectric state and the polar orientation to be relatively robust to local rotation. Such features are not found in non-uniaxial systems, such as BTO and PZT. Indeed, we emphasised the special features of LNO in the introduction of the original draft: *“LNO was chosen for several reasons: firstly, it is a uniaxial ferroelectric and hence head-to-head (H2H) or tail-to-tail (T2T) polarisation discontinuities cannot be readily obviated through 90° domain formation, or local dipolar rotation; secondly, LNO has a Curie Temperature ($\sim 1140^\circ\text{C}$) close to its melting temperature ($\sim 1240^\circ\text{C}$)¹⁹, and hence robust thermal bonding can be achieved entirely within the ferroelectric state. Uncontrolled reconstruction of the polar domains, that would inevitably occur on temperature cycling through the phase transition, is therefore avoided.”* Nevertheless, in response to the reviewer, we are content to make it clear that similar bonding using BTO or PZT as ferroelectrics of choice would be unlikely to be successful in making the same controlled interfacial polar discontinuities.

The treatment of the coincident lattice calculations in other mesh symmetries has already been done in literature (for twisted square lattices at bonded interfaces for example) and we are happy to make reference to these prior works to allow readers to see how the symmetry of the lattice distributions affects the coincident lattice parameters as a function of relative twist.

Changes to the Manuscript: We have added some text to the revised manuscript to make it clear that bonding ferroelectric crystals such as BTO and PZT would be unlikely to produce polar discontinuities. In addition, we have added some text to make it clear that the coincident lattice divergence at specific angles of twist is not unique to trigonal (0001) interfaces and added references in which other mesh symmetries have already been considered.

Comment 5: *“The cAFM measurements presented in Fig. 5D reveal conductivity at the H2H domain walls, but not at the T2T interface. To elucidate the structural origins of this conductivity, atomic-resolution HAADF-STEM should be employed to characterize atomic-scale changes across H2H domain walls. Furthermore, to enhance clarity, the authors should quantify the conductivity difference (e.g., through current histograms) and discuss whether the T2T interface exhibits any residual conductivity compared to the bulk.”*

Response 5: There are several issues to discuss here:

- (i) we should make it clear that there have already been extensive studies in which high-end electron microscopy has been used to reveal the fundamental physics associated with the emergent conductivity at H2H conventional domain walls in ferroelectrics and in LNO specifically. Such studies have resulted in three general mechanisms that can be at play: a fundamental band-gap reduction at domain walls; defect electronic states developing in the band gap at domain walls, as a result of defect aggregation; and band-bending due to the bound charge at the charged interface and associated electronic screening. In H2H walls in LNO, the consensus is that the band-bending hypothesis is probably the dominant mechanism for enhanced conductivity.
- (ii) we interpret the second half of the reviewer comment as asking for histograms of the conductivity of the H2H walls that result from polar inversion, as well as the T2T wall that then exists at the bonded interface. We think this is a good suggestion and have explicitly presented it in figure R10 below. As can be seen, there is no noticeable cAFM current signal at the bonded T2T interface. However, this was also the situation for T2T bonded interfaces that had not resulted from polar inversion. In these cases, although there were no measurable cAFM signals (see the altered figure in response to reviewer #1 comment 6), two-probe mesoscale measurements did show enhanced conductivity (see figure S8 and the response to reviewer #1 comment 7).

Changes to the Manuscript: We have added comment and references associated with previous characterisation of H2H domain walls in ferroelectrics and in LNO specifically that give strong insight into the fundamental mechanisms for emergent domain wall conduction (Schröder, M. *et al.* Conducting Domain Walls in Lithium Niobate Single Crystals. *Adv Funct Mater* **22**, 3936–3944 (2012); Nataf, G. F. *et al.* Domain-Wall Engineering and Topological Defects in Ferroelectric and Ferroelastic Materials. *Nature Reviews Physics* **2**, 634 (2020)).

Figure R10: The cAFM image associated with one of the samples in which polar inversion adjacent to the bonded interface occurred (A), along with a histogram of the current integrated over all columns in the cAFM scan, as a function of the distance away from the bonded interface. No localised cAFM signal of current exists at the T2T bonded interface, consistent with measurements on interfaces in which T2T interfaces had been deliberately engineered (rather than resulting from the local inversion in a sample that had originally been set up as H2H). We note that mesoscale 2 probe measurements at T2T bonded interfaces did show conductivity levels greater than the bulk. We expect, as mentioned several times, that the conduction at T2T interfaces is not as strongly spatially confined as in H2H cases.

References:

1. Shannon, R. D. Revised effective ionic radii and systematic studies of interatomic distances in halides and chalcogenides. *Acta Crystallographica Section A* **32**, 751–767 (1976).

2. Riefer, A. *et al.* LiNbO₃ electronic structure: Many-body interactions, spin-orbit coupling, and thermal effects. *Phys Rev B* **93**, 075205 (2016).
3. Verhoff, L. M. *et al.* Two-dimensional electronic conductivity in insulating ferroelectrics: Peculiar properties of domain walls. *Phys Rev Res* **6**, L042015 (2024).
4. Carr, S., Fang, S. & Kaxiras, E. Electronic-structure methods for twisted moiré layers. *Nat Rev Mater* **5**, 748–763 (2020).
5. Weis, R. S. & Gaylord, T. K. *Lithium Niobate: Summary of Physical Properties and Crystal Structure*. *Appl. Phys. A* vol. 37 (1985).
6. Xue, D. & Kitamura, K. Crystal Structure and Ferroelectricity of Lithium Niobate Crystals. *Ferroelectrics* **297**, 19–27 (2003).

Second Response to Reviews on NCOMMS-25-57898-T: “Polar Discontinuities, Emergent Conductivity, and Critical Twist-Angle-Dependent Behaviour at Wafer-Bonded Ferroelectric Interfaces” by Rogers *et al.*

Reviewer 1:

“The authors have demonstrated innovative twist-angle-dependent domain inversion in thermocompression-bonded lithium niobate. The extensive efforts and the amendments are highly appreciated. This manuscript is well-suited for publication in Nature Communications. The authors may consider the following constructive suggestions to further strengthen the interpretation of their results.”

Comment 1: *“The current structural evidence (e.g., Nb-column continuity in non-twisted H2H interfaces, Fig. 1, and coherent bonding at $\sim 60^\circ$ twist, Fig. 4) already provides valuable insight into interfacial strain. Nonetheless, a targeted comparison of atomic structure between H2H and T2T configurations could significantly enhance the discussion of cAFM measurement asymmetry and the mechanism of polar inversion.*

Specifically, high-resolution HAADF-STEM imaging of a non-twisted T2T interface (as likely realized in the macroscopically conducting samples of Figs. S7-S8) would be highly informative. Although Li positions remain challenging to resolve, the Nb–Nb intercolumnar spacing near the interface is readily measurable and structurally meaningful. Given the DFT prediction of opposing band-bending at H2H versus T2T junctions, one might expect expanded Nb–Nb separation in T2T.

As readers, we would like to know the structural differences between H2H and T2T via thermocompression. The relationship among twist-angle, structural change, band-bending and the polarization switching could be more elaborated. Reviewer #3 is also interested in the atomic structure in the twisted sample. This might be challenging. The non-twisted T2T interface might provide insight into it.”

Response 1: In the time available to us on SuperSTEM, we concentrated investigations on H2H interfaces as the priority. Consequently, we didn’t collect high-resolution STEM images for T2T systems. However, in the spirit of the reviewer’s comment, we decided to examine strain at T2T interfaces (no deliberate twist), inferred from HRTEM (using the FEI Talos F200X G2 system, available in Queen’s University Belfast). As can clearly be seen in figure R1 below, Geometric Phase Analysis (GPA) shows the periodicity perpendicular to the interface to be locally expanded, as suspected by the reviewer.

Figure R1: Results of Geometric Phase Analysis (GPA) on a cross-sectional HRTEM image, illustrating the strain state (ϵ_{xx}) close to a tail-to-tail interface (without deliberate relative twist). The tensile strain associated with highly localised uniaxial lattice expansion perpendicular to the interface is clear and is the opposite of the localised compression seen at the same interfaces with head-to-head polar discontinuities.

We are, however, concerned that this might be a twist-dependent phenomenon: we also compared high resolution STEM information taken on a H2H interface with $\sim 60^\circ$ twist

(data from SuperSTEM) and HRTEM on a T2T interface with $\sim 60^\circ$ twist (panels R2a and b respectively). In these datasets, GPA for the H2H sample shows clear interfacial compression close to the thermomechanical bond (panel R2c), as expected and consistent with other H2H observations at other twist angles. GPA from the corresponding T2T HRTEM is noisier (panel R2d). Nevertheless, it suggests an interfacial unit cell compression. This is surprising, given the expansion illustrated in figure R1.

So, while all observations made to date give a consistent view that the H2H bonded samples show unit cell compression perpendicular to the interface, we are concerned that the interfacial strain in T2T samples may be twist dependent. Of course, this opens up a new avenue for a much greater systematic study. However, we don't think it necessary for this publication, given that our manuscript focuses on how the screening properties of H2H bonded interfaces vary with twist.

Changes to the Manuscript: Given the lack of consistency in the nature of the strain state at the T2T bonded interfaces gleaned from GPA on HRTEM images, we think it best to restrict changes to making a brief comment in the text in the main manuscript concerning the T2T sample without a deliberate twist, as follows: *"We note parenthetically that some of our most recent data suggest this local strain state changes its sense (from compression to extension) when untwisted T2T, as opposed to H2H, polar discontinuities are considered."*

We think that formal presentation of the information contained in figures R1 and R2, outside of this response letter, would be premature, as it is based on isolated measurements and HRTEM data, as opposed to atomic resolution STEM.

Figure R2: Geometric Phase Analysis (GPA) for 60° twisted H2H and T2T bonded interfaces. (A) shows a high-resolution STEM image (SuperSTEM) from a cross-section of a twisted H2H bond; (C) shows the corresponding GPA map, with clear lattice compression along the "x" direction. (B) shows a HRTEM image of a cross-section of a twisted T2T bond. The corresponding GPA map (D) is a little noisy, but again indicates lattice compression along "x".

Comment 2: *"One final question regarding the twisted samples: in Fig. S12, the cAFM shows no signal which is consistent with the previous untwisted T2T behavior. Does the inverted T2T interface in these samples still exhibit macroscopic conductivity, as seen in Fig. S8?"*

Response 2: The reviewer is correct – neither the directly bonded untwisted T2T (Fig. S7 in the initially resubmitted manuscript), nor the twisted sample in which polar inversion has

occurred, to generate a T2T discontinuity at the bond (Fig. S12 in the initially resubmitted manuscript), shows any noticeable interfacial cAFM signal. However, since a macroscopic 2-probe signal can be seen for the untwisted one (Fig. S8 in the initially resubmitted manuscript), it would indeed be interesting to see if similar currents could be associated with the twisted and inverted T2T interface.

Unfortunately, this kind of measurement is more complicated than it might initially seem. All of our macroscopic 2-probe I-V data presented in the manuscript have been obtained using liquid InGaSn electrodes on the sample surface, into which microprobes from our probe-station can be inserted. Typically, the liquid drop has an irregular contact area, at least of the order of several hundred square microns. This makes it almost impossible to position the top electrodes onto the twisted inverted T2T interface without simultaneously contacting the nearby conventional H2H domain walls (some $10\mu\text{m}$ from the bond), which are obviously strongly conducting (as can be seen from the cAFM images) and would act electrically in parallel with the interface (thus dominating the I-V response).

Thin film patterned electrode tracks could be used to give greater spatial precision. However, the associated electrical contacts have been seen to be problematic throughout this study, decaying with time and the number of I-V cycles obtained (Fig. R3 below). Such issues do not occur when InGaSn liquid electrodes are used, and hence we have opted to use them throughout, for macroscopic transport characterisation.

Figure R3: Illustrations of frustration in making robust electrical contacts for current-voltage (I-V) measurements along the bonded interfaces. Silver dag (a,b) and sputtered gold (c,d) contacts both resulted in time and cycle-variable I-V responses. We had some hope in using a combination of the two (it was an improvement). In the end, however, the only robust mesoscale contacts found were liquid InGaSn.

Changes to the Manuscript: None.

“We emphasize that these are suggestions for improvement, not demands for infeasible tasks, such as DFT on thousands of atoms or exhaustive twist-angle mapping. A targeted T2T structural measurement would meaningfully elevate an already strong manuscript.”

Reviewer 2:

Comment 1: *“I thank the authors for their thorough and thoughtful responses to my comments. The revisions and additional analyses have significantly strengthened the manuscript. Most of my initial concerns have been adequately addressed. While a more detailed underlying physical mechanism, especially regarding the exact electronic structure modulation at twisted interfaces, remains somewhat underexplored, the phenomena reported here are highly interesting and represent an important step forward in the field of interface engineering in ferroelectrics. Given the novelty, clarity, and potential impact of the findings, I believe the manuscript in its current form is worthy of publication in Nature Communications.*

Response 1: We agree with the reviewer, insofar as follow-up work is now needed to understand exactly how short-range twist-induced aperiodicity can be responsible for the breakdown in the ability of the interface to screen positive bound charge. The suspicion is that pseudo-band gaps emerge in the electronic structure (as in quasicrystals), but this is an hypothesis which should now be more rigorously explored.

Changes to the Manuscript: None.

Reviewer 3:

Comment 1: *“I have reviewed the revised version of your manuscript and your detailed response to the reviewers' comments. I am very pleased with the revisions you have made. I have no further queries and support the acceptance of your work without any modifications.*

Response 1: Perfect.

Changes to the Manuscript: None.

Third Response to Reviews on NCOMMS-25-57898-T: “Polar Discontinuities, Emergent Conductivity, and Critical Twist-Angle-Dependent Behaviour at Wafer-Bonded Ferroelectric Interfaces” by Rogers *et al.*

Reviewer 1:

Comment 1: *“I would like to thank the authors for their detailed and thoughtful responses to my comments. I appreciate the additional efforts made to characterize the T2T interfaces using HRTEM with GPA, despite the limited availability of SuperSTEM time. Most of my concerns have been adequately addressed. Although the structural information is still under exploration, which is one of the keys to elucidating a more detailed physical mechanism, the findings of interfacial ferroelectric engineering demonstrate high novelty. I have no further suggestions and support the acceptance of the revised manuscript.”*

Response 1: We thank the reviewer for their comments throughout the review process and are happy that they now support the acceptance of our manuscript.